# IKK2 controls the inflammatory potential of tissue-resident regulatory T cells in a murine gain of function model

Chelisa Cardinez[1,2,3,4], Yuwei Hao [1,2,3,5], Kristy Kwong [1,2,3,5], Ainsley R. Davies [1,2,3], Morgan B. Downes [1,2,3], Nadia A. Roberts [3], Jason D. Price[4], Raquel A. Hernandez[1,2,3], Jessica Lovell[3], Rochna Chand[1,2,3], Zhi-Ping Feng [6], Anselm Enders [1,3], Carola G. Vinuesa [1,7], Bahar Miraghazadeh[1,2,3] & Matthew C. Cook [1,2,3,5] ✉

Loss-of-function mutations have provided crucial insights into the immunoregulatory actions of Foxp3+ regulatory T cells (Tregs). By contrast, we know very little about the consequences of defects that amplify aspects of Treg function or differentiation. Here we show that mice heterozygous for an *Ikbkb* gain-of-function mutation develop psoriasis. Doubling the gene dose (*Ikbkb*^GoF/GoF) results in dactylitis, spondylitis, and characteristic nail changes, which are features of psoriatic arthritis. *Ikbkb*^GoF mice exhibit a selective expansion of Foxp3 + CD25+ Tregs of which a subset express IL-17. These modified Tregs are enriched in both inflamed tissues, blood and spleen, and their transfer is sufficient to induce disease without conventional T cells. Single-cell transcriptional and phenotyping analyses of isolated Tregs reveal expansion of non-lymphoid tissue (tissue-resident) Tregs expressing Th17-related genes, Helios, tissue-resident markers including CD103 and CD69, and a prominent NF-κB transcriptome. Thus, IKK2 regulates tissue-resident Treg differentiation, and overactivity drives dose-dependent skin and systemic inflammation.

Foxp3+ regulatory CD4+ T cells (Treg) are typically suppressor cells that mediate dominant tolerance, which is crucial for the prevention of organ-specific autoimmune disease[1–4]. During their development in the thymus, Tregs undergo positive selection mediated by high-avidity interactions between T cell receptors (TCR) and self-antigen[5]. Helios marks cells that have registered a strong TCR signal during single positive CD4+ T cell differentiation and are therefore poised for either Treg formation or deletion[6]. Helios+ Foxp3- Treg precursors are rescued from the alternative fate of apoptosis by IL-2, which promotes Foxp3 expression[7]. Since Treg selection depends on a strong TCR

signal, it follows that Treg development requires competent signalling downstream of the TCR. Signalling via the CARD11-BCL10-MALT1 (CBM) complex that links TCR ligation to canonical NF-κB activation appears to be crucial for preventing apoptosis after strong TCR signals. Mice deficient in PKCδ, CARD11, BCL10 as well as IKK2 or c-Rel all exhibit Treg deficiency[8–16]. Specifically, CBM and NF-κB defects result in a selective loss of CD4+ Helios+ thymic T cells that have not yet upregulated Foxp3.

After maturation in the thymus, Tregs migrate between secondary lymphoid organs, based on expression of adhesion molecules,

[1]Centre for Personalised Immunology, John Curtin School of Medical Research, Australian National University, Canberra, ACT, Australia. [2]Translational Research Unit, The Canberra Hospital, Canberra, ACT, Australia. [3]Division of Immunology and Infectious Diseases, John Curtin School of Medical Research, Australian National University, Canberra, ACT, Australia. [4]Division of Genome Sciences and Cancer, John Curtin School of Medical Research, Australian National University, Canberra, ACT, Australia. [5]Cambridge Institute of Therapeutic Immunology and Infectious Disease, Department of Medicine, University of Cambridge, Cambridge, UK. [6]ANU Bioinformatics Consultancy, John Curtin School of Medical Research, Australian National University, Canberra, ACT, Australia. [7]Francis Crick Institute, London, UK. ✉e-mail: mc2386@cam.ac.uk

including CD62L and CCR7. A subset of recirculating Tregs exhibit an activated or effector phenotype (eTreg) marked by expression of BLIMP1 and CD38, and downregulation of CCR6, ICOS and Bcl2[17,18]. Transcriptional analysis has also identified Treg subsets adapted to specific regulation of Th1, Th2, Th17 and Tfh responses[19–23]. These subsets co-opt parts of the transcriptional programmes of the corresponding effectors, activated in response to similar cytokine cues that drive differentiation of their effector counterparts. This set-up provides a mechanism by which immune regulation is built in to inflammatory responses within particular microenvironments and in response to a cluster of antigens but also means there is an inherent risk of immune disease if eTreg differentiation results in adoption of an overt effector phenotype. Under most circumstances, persistent expression of Foxp3 renders Tregs robust to this outcome, although there is some evidence that Tregs may indeed adopt effector function[24]. At present the mechanisms restraining this conversion and any implications for pathology remain unknown.

Tregs can differentiate further and become tissue-resident cells in non-lymphoid parenchyma (non-lymphoid tissue Tregs, NLT), where they are thought to contribute not only to immune regulation, but also tissue homeostasis[25–27]. scRNASeq has identified possible functional mediators of NLT including *Areg, Nrp1, Il1rl1, Ikzf2*, and differences in Tregs retrieved from different end-organs, including a prominent TNF-NF-κB signature in NLT[28].

Our understanding of immune regulation by Tregs has been advanced significantly by gene deletion (loss-of-function, LoF) studies[29]. Much less attention has been given to elucidating the cellular and molecular pathways of gain-of-function (GoF) mutations. By contrast, analysis of rare human inborn errors of immunity has revealed many pathological GoF variants, and the resulting phenotypes are not necessarily predictable based on LoF variants of the same genes[30]. Since the genetic architecture of non-Mendelian immune disease is likely to encompass genetic variants of both polarities, analysis of strong acting GoF mutations provides an important avenue for understanding immune regulation and disease pathogenesis.

Heterozygous GoF mutations in *IKBKB* (encoding IKK2) result in combined immune deficiency, while homozygous LoF results in severe combined immune deficiency[31–34]. In order to investigate the mechanism of action of the GoF mutation, we engineered mice to carry the orthologous genetic variant. Heterozygous mice developed skin disease with histological characteristics of psoriasis and doubling the gene dose of the GoF allele (i.e. homozygosity) resulted in an accurate model of psoriatic arthritis. These *Ikbkb*mut mice exhibit a gene dose-dependent expansion of Tregs at the inflamed tissues and spleen, of which a subset express IL-17. While these findings appear paradoxical, further analysis revealed induction of inflammation after transfer of these Tregs. By single-cell transcriptome and phenotyping analyses, this population as a modified subset of skin NLT Tregs is characterised by NF-κB activity, Th17-related genes, Helios and tissue-resident markers. Our findings reveal that a subset of NLT can be driven to effector-like function by excess NF-κB activity, which perturbs their homeostatic action to drive pathology.

## Results

### Overactive IKK2 leads to accumulation of dermal Tregs and psoriasis

We analysed mice bearing an *Ikbkb* GoF mutation orthologous to a pathological human *IKBKB* GoF variant (p.Val203Ile)[31]. We observed that both heterozygous (*Ikbkb*mut/+) and homozygous (*Ikbkb*mut/mut) mutant mice developed dermatitis with age, including thickened and macroscopically shortened tails (Fig. 1a; Fig. S1a). We followed a cohort of mice of different *Ikbkb* genotypes and documented the age of onset of skin pathology. Abnormal *Ikbkb*mut/+ and *Ikbkb*mut/mut mice were flagged by animal house technicians blinded to mouse genotypes, and this revealed that *Ikbkb*mut/mut mice developed a pathological tail

phenotype as early as 30 days of age. By 120 days, all mice were affected (Fig. 1b). Heterozygous mice also developed this phenotype, although with incomplete penetrance, so that even at 190 days more than 50% of mice remained unaffected. Histological analysis undertaken by a pathologist blinded to mouse genotypes revealed evidence of patchy inflammation of the skin and hair follicles in *Ikbkb*mut/+ mice (Fig. 1c). These changes were most marked in *Ikbkb*mut/mut mice and included dermal inflammation with a mononuclear cell infiltrate and epidermal acanthosis. We observed a 2-fold change in average dermal thickness in the *Ikbkb*mut/mut group and a less marked but significant increase in epidermal thickness (Fig. 1d, e). *Ikbkb*mut/mut mice also developed ear dermatitis (Fig. 1f), which developed later than the tail changes and exhibited incomplete penetrance. 65% of *Ikbkb*mut/mut mice developed ear dermatitis, while 95% of the heterozygous cohort remained dermatitis-free at 200 days (Fig. 1g).

Similar to changes observed in the tails, histological analysis of the ears also revealed features of psoriasis with dermal and epidermal thickening, acanthosis and hyperkeratosis (Fig. 1h). *Ikbkb*mut/mut mice exhibited epidermal collections of neutrophils reminiscent of Munro's microabscesses, which are a highly specific histopathological sign of psoriasis[35]. There was a 3-fold increase in thickness of both dermis and epidermis in the ears of *Ikbkb*mut/mut mice (Fig. 1i, j).

Skin analysis by histology revealed hallmarks of psoriasis. In order to confirm the pattern of inflammation, we compared transcriptomes of tails and ears from *Ikbkb*mut/mut and *Ikbkb*+/+ mice. Human psoriasis is characterised by Th17/1-related skin gene signatures. IL-23 antagonism is therapeutic in psoriasis, confirming the importance of IL-23 in the persistence of psoriatic lesions[36]. By contrast, atopic dermatitis tissues show preferential Th2 skewing[37]. Tails and ears from affected mice both showed transcriptional signatures consistent with psoriasis. We observed significant upregulation of *Il23* in ear and tail tissues from *Ikbkb*mut/mut mice. Furthermore, the transcriptional signature from affected skin revealed upregulation of key human psoriasis genes, including *Lcn2* and *Defb4*, cytokines *Tnf, Il1b*, and *Il36a/g*, and chemokine ligands *Ccl2, Cxcl9*, and *Ccl20* (Fig. 1k, Supplementary data 1-2). By contrast, atopic dermatitis and Th2-related genes, including *Il4, Il5, Il13* and *Il31* were lowly expressed or not detected at all, and other atopic dermatitis-related transcripts (*Ccr4, Ccl17, Ccl24*) were not significantly differentially expressed. Thus, transcriptional analysis of mouse skin confirmed the fidelity of *Ikbkb*GoF as a model of human psoriasis.

Inflammation of back skin was also observed in mutant mice, although this occurred later and with reduced penetrance than ear and tail inflammation and was sometimes only evident after removing the hair (Fig. S1b). We investigated the inflammatory infiltrate in skin lesions. Cells were recovered from digested skin tissue from ears, tails, and back. Total cell infiltrates (measured as cells/g of tissue) were higher in mutant mice relative to wild type for each skin site (Fig. S1c–e). Leukocytes in the tail accounted for a higher proportion of cells recovered from mutant mice (Fig. S1f, g). The expansion of lymphocytes was accounted for by increased αβ CD4+ T cells and to a lesser extent αβ CD8+ T cells, whereas there was a relative reduction of γδ T cells (Fig. S1h–l). Interestingly, we observed a significant expansion of CD25+ Foxp3+ Tregs as a proportion of CD4+ T cells in the back and tail skin of heterozygous and homozygous mice relative to WT (Fig. 1l–n). Additionally, Foxp3+ Treg counts were higher in mutant mice relative to wild type in back and tail skin (Fig. S1m, n).

### Double dose of *Ikbkb*mut results in systemic inflammation and psoriatic arthritis

Further observation of *Ikbkb*mut mice revealed that in addition to dermatitis, homozygotes went on to develop arthritis, which was concentrated in the digits of all paws (Fig. 2a, b) but also affected ankles of many mice (Fig. 2b). Digital arthritis occurred universally in mice homozygous for *Ikbkb*mut but did not develop in heterozygotes despite

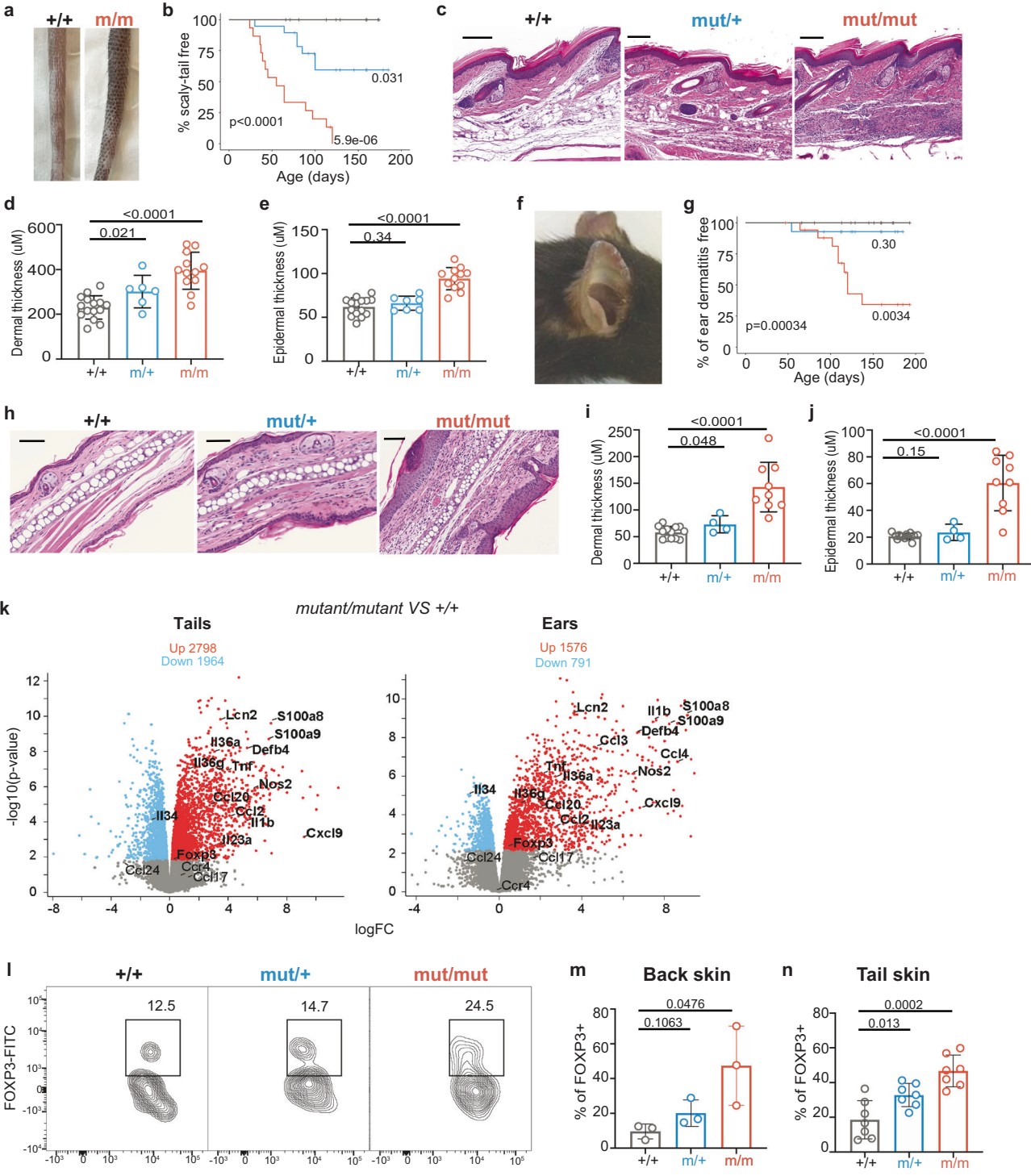

**Fig. 1 | Overactive IKK2 leads to psoriasis.** Representative images (**a**) and Kaplan-Meier plots of cumulative incidence (**b**) of tail pathology in *Ikbkb*^+/+ (*n* = 13), *Ikbkb*^mut/+ (*n* = 19) and *Ikbkb*^mut/mut (*n* = 15) mice. Tail skin pathology revealed by representative images of skin sections stained with H&E (scale bar, 200 μm) (**c**), and summaries of dermal (**d**) and epidermal thicknesses (**e**). *Ikbkb*^+/+, *n* = 15; *Ikbkb*^mut/+, *n* = 7, *Ikbkb*^mut/mut, *n* = 12. Ear skin pathology revealed by a representative image from an *Ikbkb*^mut/mut mouse (**f**), Kaplan-Meier plots of cumulative incidence of dermatitis by genotype (**g**) (*n* = 13-18/genotype), representative images of sections of ear skin stained with H&E (scale bar, 100 μm) (**h**), and summaries of dermal (**i**) and epidermal thicknesses (**j**). *Ikbkb*^+/+, *n* = 15; *Ikbkb*^mut/+, *n* = 4; *Ikbkb*^mut/mut, *n* = 9. **k** Volcano plot depicting differentially expressed genes in *Ikbkb*^mut/mut and *Ikbkb*^+/+ mice from

tail and ear skin (*n* = 3/genotype). Red dots represent genes highly expressed in mutants relative to wild type while blue dots represent genes lowly expressed in mutants relative to wild type. Gene expression was normalized using trimmed mean of M values and differentially expressed genes were identified with a Benjamini−Hochberg adjusted *p* value < 0.05. Flow cytometric analysis of Foxp3 expression by CD4+ T cells from skin for indicated *Ikbkb* genotypes, with representative plots (**l**), enumeration of Tregs as a proportion of CD4+ T cells from back skin (*n* = 3/genotype) (**m**) and tail skin (*n* = 7/genotype) (**n**). Summary graphs are presented as mean +/− s.d. Kaplan-Meier curves: log-rank (Mantel-Cox) test; Frequency histograms, one-way ANOVA with Bonferroni's multiple comparison test.

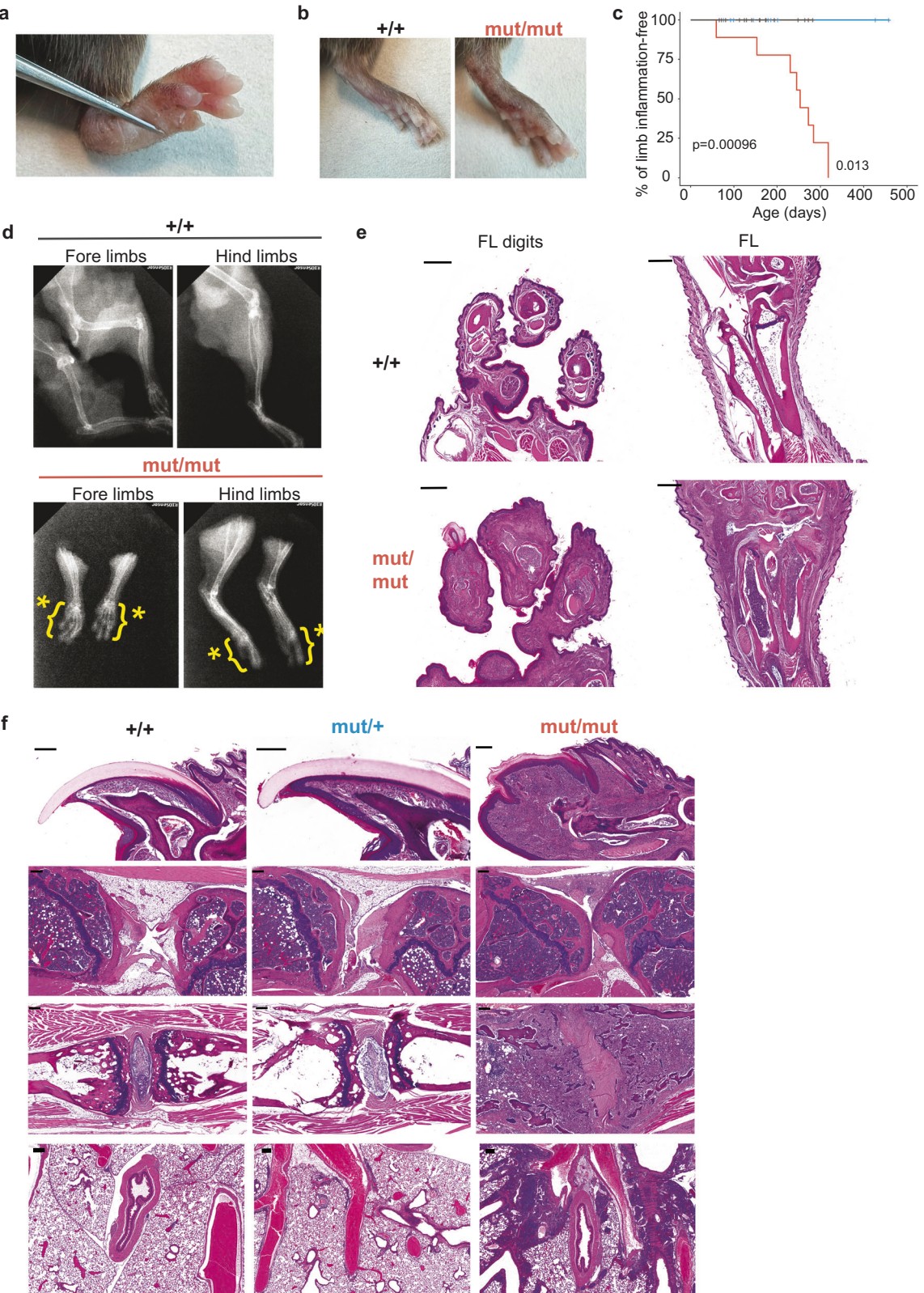

**Fig. 2 | Double dose of *Ikbkb*<sup>mut</sup> results in systemic inflammation and psoriatic arthritis.** Arthritis shown by representative images of dactylitis in an *Ikbkb*<sup>mut/mut</sup> mouse compared with a *Ikbkb*<sup>+/+</sup> control (**a**, **b**) and Kaplan-Meier plots of cumulative incidence of limb arthritis by genotype (**c**). *Ikbkb*<sup>+/+</sup>, n = 32; *Ikbkb*<sup>mut/mut</sup>, n = 11; *Ikbkb*<sup>mut/+</sup>, n = 17. **d**. Representative radiographs of limbs from *Ikbkb*<sup>+/+</sup> and *Ikbkb*<sup>mut/mut</sup> mice. Asterisk indicates abnormal skeletal structure. **e** Representative images of fore limb (FL) digits and long bones stained by H&E. **f** Histological survey of mice by genotype showing sections of nails (*top row*), ankles (*second row*), tail vertebrae (*third row*), and lungs (*fourth row*) all stained by H&E. Scale bars, 200 µm. *Ikbkb*<sup>+/+</sup>, n = 3; *Ikbkb*<sup>mut/+</sup>, n = 4; *Ikbkb*<sup>mut/mut</sup>, n = 3 (**d**, **f**). Statistical analysis of Kaplan-Meier plots by log-rank (Mantel-Cox) test.

observation out to 500 days of age (Fig. 2c). Radiological examination revealed bone and joint deformities in both fore and hind limbs of *Ikbkb*<sup>mut/mut</sup> mice, most notably in the phalanges (dactylitis) (Fig. 2d). Histological analysis of the paws confirmed the presence of dactylitis with bone marrow expansion and inflammation in digital soft tissues (Fig. 2e). Histological analysis of the feet showed pronounced enlargement and swelling of the digits with severe inflammation in and around the distal joints of all limbs, characterised by the presence of pleomorphic mononuclear and neutrophilic infiltrates (Fig. 2e).

Nail changes are a hallmark of psoriatic arthritis[38]. *Ikbkb*<sup>mut/mut</sup> mice developed florid structural abnormalities in their nails (Fig. 2f). These were obvious macroscopically. Histological analysis revealed thickening of the nail bed with mononuclear and neutrophilic infiltrates, contiguous with dermal infiltrates of skin and foot pads. In humans, nail changes correlate with development of psoriatic arthritis. Similarly, in the mouse model, nail changes occurred in homozygotes and not in *Ikbkb*<sup>mut/+</sup> mice.

Post-mortem examination revealed that *Ikbkb*<sup>mut/mut</sup> mice developed bone marrow expansion in proximity to peripheral joints (Fig. 2f). *Ikbkb*<sup>mut/mut</sup> mice had spondyloarthropathy, with prominent bone marrow expansion in vertebral bodies. Examination of bone marrow from affected mice revealed a selective expansion of CD4+ T cells, with no increase of either B cells or CD8+T cells in mutant mice (Fig. S2a–d). We observed a significant increase in Foxp3+ CD4+ T cells in bone marrow of affected mice (Fig. S2e, f).

## IL-17-producing Tregs are found in *Ikbkb*<sup>mut</sup> mice

So far, we have described a gene-dose dependent development of psoriatic pathology conferred by increased IKK2 activity, and the paradoxical expansion of Tregs within the pathological lesions. Next we investigated the nature of this Treg expansion. As we reported previously, humans heterozygous for *IKBKB*<sup>V203I</sup> exhibit combined immune deficiency but with expansion of Tregs[31]. *Ikbkb*<sup>mut</sup> exhibit a similar phenotype, with gene dose-dependent lymphopenia (Fig. 3a), accounted for by a reduction of both αβ and γδ T cells in homozygous mice (Fig. S3a–c). Consistent with previous findings, analysis of splenocytes and pooled lymph nodes revealed a reduction of CD3+ T cells, more marked in the CD8 than CD4 compartments (Fig. S3d–g)[31]. Heterozygous mice exhibited a phenotype intermediate between homozygous mutant and WT mice (Fig. 3a, S3a–c). Tregs were increased as a proportion of CD4+ T cells in spleen, pooled lymph nodes, blood and thymus of *Ikbkb*<sup>mut/+</sup> and *Ikbkb*<sup>mut/mut</sup> mice (Fig. S3h–k). In mutant mice, the majority of Tregs co-expressed Helios, whereas Helios+ Tregs are a minority in WT mice (Fig. 3b, c). There is some evidence to suggest that Helios expression indicates thymic Tregs[39,40] and we also observed a relative expansion of Tregs in the thymus (Fig. S3k). In addition, we observed that a higher proportion of mutant splenic Tregs were Ki67+ (Fig. S3m, n). This indicates that the mutant Treg population has increased ability to proliferate compared to a Treg population from *Ikbkb*<sup>+/+</sup> mice.

To test whether the Treg expansion was a consequence of a cell-intrinsic action of the *Ikbkb* mutation, we generated mixed bone marrow chimaeras using allotype marked donor cells from WT and mutant mice (Fig. 3d–f). At 8 weeks, we observed a gene dose-dependent expansion of allotype-marked mutant cells, while the proportion of WT cells was similar irrespective of the presence or absence of mutant cells (Fig. 3f).

Our index patient also exhibited an expansion of Th17 CD4+ T cells (Fig. S4a). Human *IKBKB*<sup>GoF</sup> appears to be very rare but the index patient also presented with acne inversa (hidradenitis suppurativa), and another with acne congoblata, both IL-17-related inflammatory conditions that are significantly associated with psoriasis[31,32,41–43]. We observed a similar expansion of IL-17+ CD4+ T cells in *Ikbkb* mutant mice where a gene-dose effect was evident, with greater expansion observed in *Ikbkb*<sup>mut/mut</sup> mice than heterozygotes (Fig. 3g, h). In order to determine if this bias to IL-17A+ CD4+ T cells arose from a cell-intrinsic action of IKK2, we purified naïve (CD44− CD62L+) CD4+ T cells from whole mouse splenocyte suspensions and stimulated them in vitro under Th17-inducing conditions. Surprisingly, we observed that the proportion of Th17 cells arising from *Ikbkb*<sup>mut/mut</sup> cells was significantly reduced when compared with *Ikbkb*<sup>mut /+</sup> and *Ikbkb*<sup>+/+</sup> cells (Fig. 3i, j), providing evidence against a CD4+ T cell-intrinsic bias towards Th17 effectors. Similarly, when conventional CD4+ T cells were stimulated under Th1-inducing conditions, there was no evidence of a cell-intrinsic bias towards Th1 differentiation in mutant cells (Fig. S4b, c). Collectively, these results provided evidence that conventional CD4+ T cells are not the source of IL-17 in *Ikbkb* mutant mice.

Since we had observed substantial expansion of Tregs with *Ikbkb*<sup>mut</sup>, we investigated whether this subset might account for the observed expansion of IL-17+ CD4+ T cells. To this end, we enumerated IL-17+ Tregs ex vivo. Splenocytes were stimulated for 6 hours with PMA and Ionomycin and then stained for cytokine production after gating on Foxp3. This revealed increased IL-17 production by Tregs in the spleen from *Ikbkb*<sup>mut/mut</sup> mice relative to WT (Fig. 3k, l). Expansion of IL-17+ Tregs were also evident in blood of mutant mice (Fig. S4d). IFNγ production by Tregs appeared to be similar between WT and mutant mice (Fig. 3m). These findings indicate that *Ikbkb*<sup>mut</sup> confers an expansion of IL-17-producing Foxp3+ Treg population. Further analysis of the lesional Tregs revealed gene-dose dependent expansion of IL-17+ Foxp3+ CD4+ T cells retrieved from the skin (Fig. 3n-o) whereas there was no increase in IFNγ+ Tregs (Fig. 3p). Alternatively, gating on IL-17+ CD3+ cells revealed that the majority expressed Foxp3 (Fig. 3q, r).

## Foxp3 + CD4 + T cells from *Ikbkb*<sup>mut</sup> mice retain suppressive function

Having observed this unusual phenotype of Tregs in *Ikbkb*<sup>mut</sup> mice, we proceeded to investigate their conventional immunosuppressive function. First, we isolated Tregs from WT mice as well as mice either heterozygous or homozygous for *Ikbkb*<sup>mut</sup> and cocultured them with purified WT conventional T cells labelled with CTV, either in a 1:1 ratio, or a 3:1 excess of Tregs. Cocultures were stimulated with CD3 and CD28 and proliferation determined by dilution of CTV on day 3. We did not identify any significant difference in the ability of mutant and WT Tregs to suppress proliferation in vitro (Fig. 4a, b).

Next, we performed a similar test of mutant Treg suppression but this time in vivo. To facilitate purification of unmanipulated Tregs, we crossed WT, *Ikbkb*<sup>mut/+</sup> and *Ikbkb*<sup>mut/mut</sup> mice with Foxp3-GFP transgenics and isolated Tregs according to GFP expression. Tregs were transferred with WT conventional CD4+ T cells into *Rag1*<sup>−/−</sup> recipients (Fig. 4c). Using this approach, absence of Treg suppression results in colitis, which is detectable as weight loss, and specifically through histological changes in the colon (Fig. 4e, f). We observed a mild but significant phenotype consistent with a modest Treg defect in recipients of *Ikbkb*<sup>mut/mut</sup> Tregs (Fig. 4d, f). These findings suggest that mutant Tregs maintain some suppressor function although this may be less robust than in WT mice.

Next we examined mice for evidence of generalised immune dysregulation. Foxp3 deficiency and Treg functional defects are associated with early onset and marked generalised lymphadenopathy[44–46]. This phenotype is not conferred by the *Ikbkb*<sup>mut</sup> mutation, as mesenteric lymph nodes were similar in size in both mutants and *Ikbkb*<sup>+/+</sup> mice (Fig. S5a, b). Mesenteric lymph nodes drain the colon and small intestine where we found no evidence of spontaneous inflammation (Fig. S5c). In humans, lymphadenopathy is observed in rheumatoid arthritis, juvenile idiopathic arthritis, and ankylosing spondylitis[47], where it occurs in lymph nodes draining inflamed joints. Interestingly, we observed inguinal, brachial, and cervical lymphadenopathy in mutant mice, more marked in *Ikbkb*<sup>mut/mut</sup> than *Ikbkb*<sup>mut/+</sup> mice (Fig. S5d–f). Thus, lymphadenopathy appears to be located at sites draining tissue-specific inflammation, and thus reactive rather than a

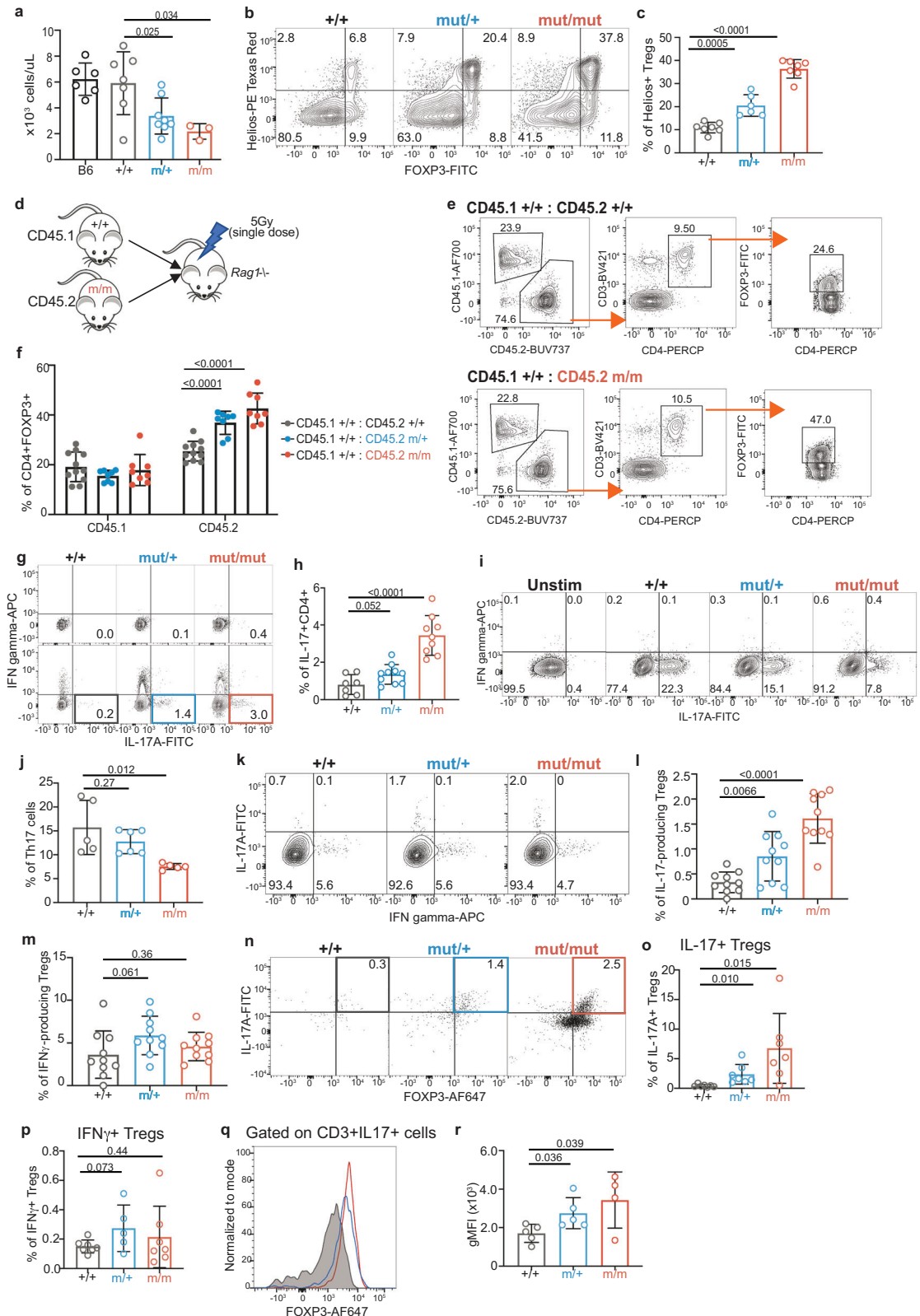

consequence of generalised lymphoproliferation. Absence of inflammation in the gastrointestinal tract represents further evidence against Treg deficiency as the cause of pathology.

**Treg expansion results from a T cell-intrinsic action of *Ikbkb*[mut]**
Components of the canonical NF-κB pathway are expressed widely, including in cells of the immune system and skin. Furthermore,

conditional deletion of *Ikbkb* from epithelium has been shown to induce changes in immune cells[48]. In order to investigate cell-intrinsic actions of *Ikbkb*[mut] on the Treg phenotype, we generated reciprocal bone marrow (BM) chimeras such that the *Ikbkb* mutation was confined to either the hematopoietic or non-hematopoietic tissues (Fig. 5a). Analysis of circulating lymphocytes at nine weeks revealed adequate reconstitution of all mice. We observed relative expansion of

**Fig. 3 | IL-17-producing Treg population arises in mice with an *Ikbkb* GoF variant. a** Summary of lymphocyte counts from spleens from *Ikbkb* mice and C57Bl/6 controls. C57Bl/6, *n* = 6; *Ikbkb*^+/+^, *n* = 7; *Ikbkb*^mut/+^, *n* = 8; *Ikbkb*^mut/mut^, *n* = 3. Flow cytometric analysis of Helios and Foxp3 expression by splenic CD4+ T cells for each genotype, with representative plots (**b**) and a summary of the proportion of Helios+ Foxp3+ Tregs by *Ikbkb* genotype (**c**). *Ikbkb*^+/+^, *n* = 7; *Ikbkb*^mut/+^, *n* = 6; *Ikbkb*^mut/mut^, *n* = 7. Analysis of mixed bone marrow chimeras constructed, showing the experimental design (**d**), gating strategy for allotype-marked splenic Foxp3+ CD4+ Treg cells (**e**), and results for the Foxp3+ Treg cells expressed as proportions of splenocytes (**f**). *Ikbkb*^+/+^, *n* = 11;, *Ikbkb*^mut/+^, *n* = 8, *Ikbkb*^mut/mut^, *n* = 8 from two independent experiments. Intracellular expression of IFNγ and IL-17A by CD4+ T cells by *Ikbkb* genotype, showing representative plots (**g**) and summary results (**h**). *Ikbkb*^+/+^, *n* = 7;, *Ikbkb*^mut/+^, *n* = 10, *Ikbkb*^mut/mut^, *n* = 9. Induction of IFNγ and IL-17A expression after culturing naïve CD4+ T cells under Th17-inducing conditions showing representative flow cytometric analysis (**i**) and summary results (**j**) (*n* = 5–6 biological replicates/genotype). Ex vivo expression of IL-17A (**k**, **l**) and IFNγ (**k**, **m**) by splenic Foxp3 + CD4+ T cells, showing representative flow cytometric analysis (**k**) and summary results (**l**, **m**), *n* = 10/genotype. Analysis of IL-17A and Foxp3 expression by CD4+ cells recovered from tail skin, showing representative flow cytometric analysis (**n**) and summaries of IL-17A⁺ (**o**) and IFNγ⁺ (**p**) Foxp3⁺ lymphocytes as a proportion of CD45+ cells, *n* = 6-7/genotype. **q** Representative flow cytometric histograms of Foxp3 expression by CD3 + IL-17A+ cells from skin. **r** Summary of geometric mean fluorescence intensity (gMFI) of Foxp3 expression in IL-17A + CD3+ cells. Summary graphs are from 2-3 independent experiments and show mean +/– s.d. Tests of statistical significance were performed by one-way ANOVA with Bonferroni's multiple comparison test. Each symbol represents a biological replicate.

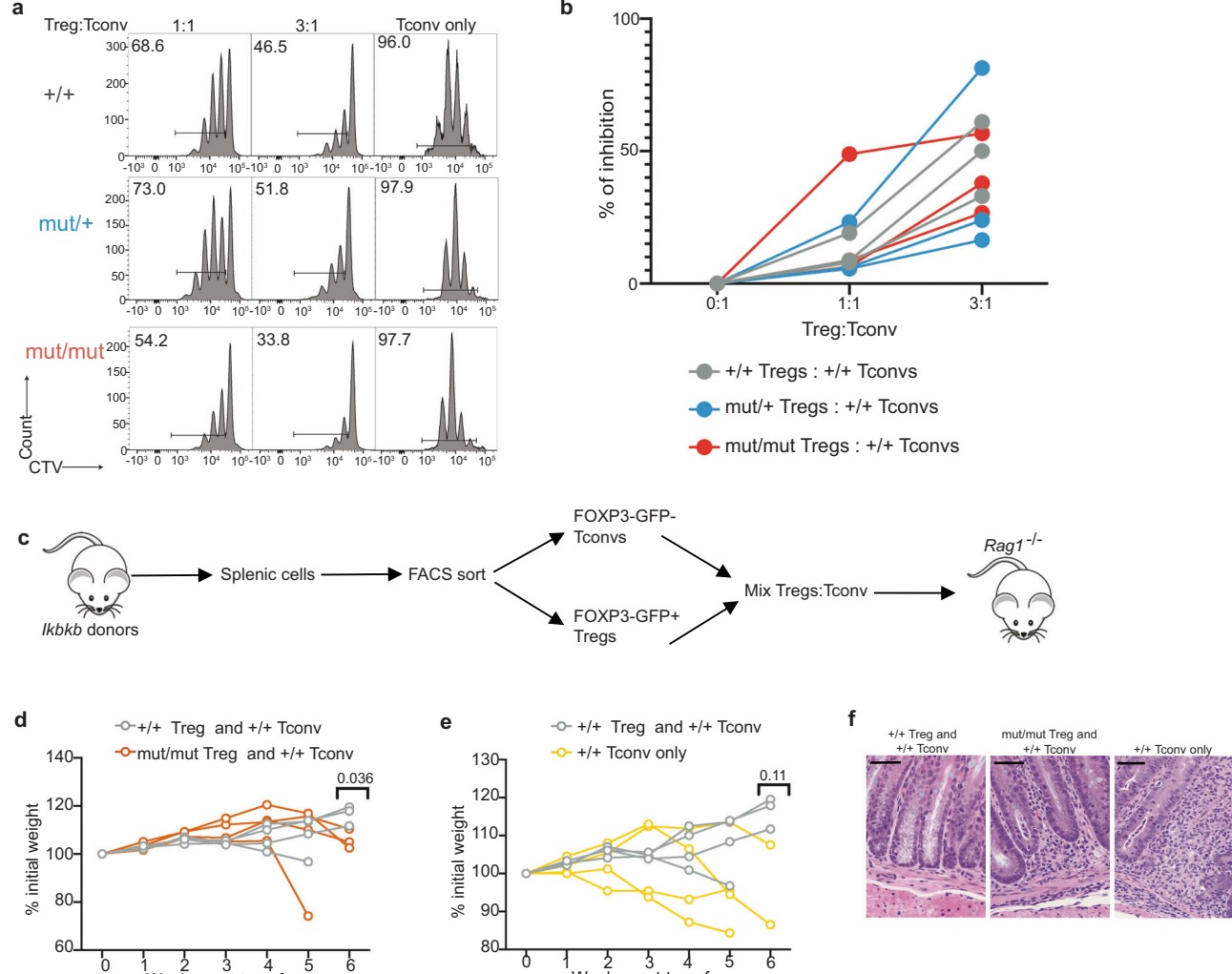

**Fig. 4 | Foxp3 + CD4+ T cells retain suppressive function. a, b** Flow cytometric evaluation of conventional T cell proliferation by CTV dilution during coculture with Tregs. Representative histograms indicate the proportion of live cells that have completed >1 division for each Treg genotype (**a**) and summary of inhibition (**b**). Representative of 2 independent experiments. Adoptive transfer experiment showing experimental design (**c**), change in body mass after adoptive transfer of cells from indicated strains (**d**, **e**). Each line represents results from one mouse, *n* = 3-4 per group (defined as shown). Comparison at week 6 post-transfer by two-tailed unpaired t-tests. **f** Representative histology of colon samples, H&E staining. Scale bars, 50 μm. *n* = 4 per group.

Foxp3+ Tregs in recipients of *Ikbkb*^mut^ BM, while in *Ikbkb*^mut^ recipients of WT BM, the proportion of CD4+ T cells that expressed Foxp3 was similar to the proportion observed in intact WT mice (Fig. 5b). We collected serum from recipient mice for analysis of a panel of cytokines, which revealed a significant increase of IL-17A in recipients of mutant BM (Fig. 5c). We also observed an increase in TNF and IFNγ (Fig. 5c), while IL-10, IL-5, IL-2, IL-33 and IL-6 were similar in both sets (Fig. S6). The following cytokines: IFNα, IFNβ, IL-12p70, IL-1b and IL-4 were also measured but were found to be below lower limit of detection for the assay. While macroscopic evidence of arthritis was not obvious in either group of chimeras, post-mortem analysis revealed significant peripheral and axial arthritis in WT recipients of *Ikbkb*^mut^ BM

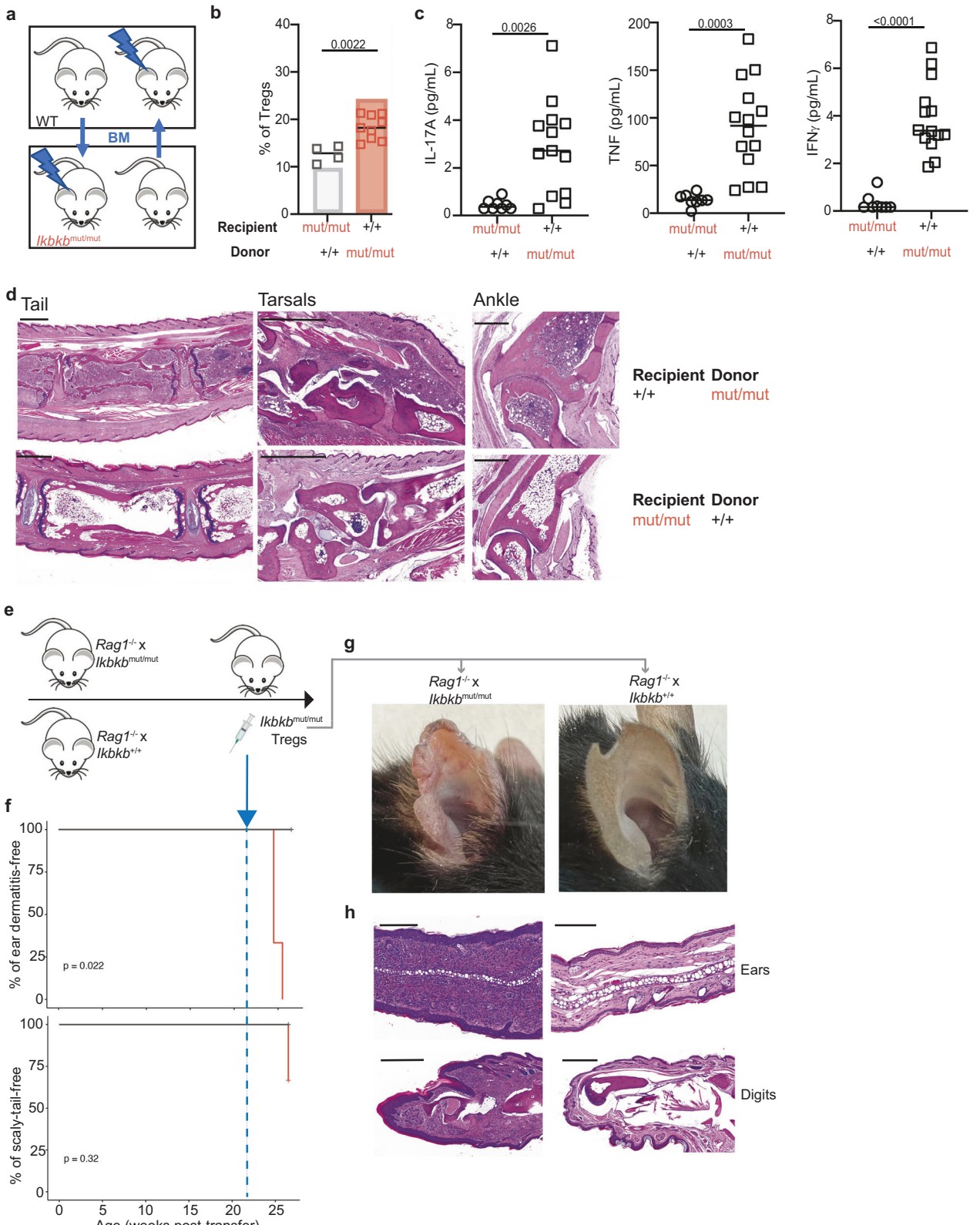

**Fig. 5 | Treg expansion results from a T cell-intrinsic action of *Ikbkb*^mut.** Reciprocal bone marrow chimera experiment (*Ikbkb*^mut/mut^ recipients, $n = 4$; *Ikbkb*^+/+^ recipients, $n = 9$), showing experimental design (**a**), proportions of CD25+Foxp3+ Tregs in indicated chimeric strains (**b**), serum IL-17A, TNFα and IFNγ concentrations (**c**) (*p* values from Student's t-tests), and representative histology of tail, tarsals, and ankles of chimeras (scale bars, 1000 μm for tails and tarsals, 500 μm for ankles) (**d**). The transparent bar in **b** shows mean Treg proportions observed in intact *Ikbkb*^+/+^

mice (grey) and *Ikbkb*^mut/mut^ mice (red) for comparison (as shown in Fig. 1e). Adoptive transfer experiment showing experimental design (**e**), Kaplan-Meier plots of mice remaining free of ear dermatitis after adoptive transfer of sorted *Ikbkb*^mut/mut^ x *Foxp3*-GFP Tregs (**f**). Representative images (**g**) and sections of ears and digits stained by H&E after second phase transfer of *Ikbkb*^mut/mut^ Tregs (**h**). Scale bars, ears, 200 μm; digits, 500 μm. $n = 3$ per group. Comparison of Kaplan Meier curves was performed using a log-rank test.

(Fig. 5d), which was not evident in the reciprocal mice. Inflammation was particularly marked in tarsal bones and spine.

In order to investigate the action of *Ikbkb*mut in non-lymphoid tissue, we crossed *Ikbkb*mut mice onto a *Rag1*−/− background and followed them for signs of pathology. *Ikbkb*mut x *Rag1*−/− and *Ikbkb*WT x *Rag1*−/− remained indistinguishable and healthy out to 21 weeks (Fig. 5e, f). Next, to confirm that pathology was conferred by a pro-inflammatory action of Tregs (rather than defective suppression), we purified GFP-labelled Foxp3+ Tregs (as described above) from intact *Ikbkb*mut donors and injected them as a second phase experiment into either *Ikbkb*mut x *Rag1*−/− or *Ikbkb*WT x *Rag1*−/− mice. We observed development of skin disease within three weeks of transfer of mutant Tregs into *Ikbkb*mut x *Rag1*−/− recipients but not in *Ikbkb*WT x *Rag1*−/− recipients (Fig. 5g). Histological analysis confirmed development of pathology in recipients of mutant Tregs in ears and digits (Fig. 5h). Together, these findings reveal that mutant Tregs are necessary but not sufficient to cause pathology in the absence of *Ikbkb*mut outside the haematopoietic system. In particular, systemic inflammation is driven by mutant lymphocytes. Furthermore, inflammation in this model is driven by the proinflammatory action of mutant Foxp3+ Tregs rather than a defect in Treg-mediated suppression.

### Pathogenic Tregs are modified non-lymphoid tissue skin Tregs that are NF-κB-dependent

So far, our results suggest that *Ikbkb*GoF induces the formation of pathogenic IL-17+ Tregs but not all Tregs in this model acquire an IL-17+ phenotype. To resolve this heterogeneity, we proceed to analyse the Foxp3+ Treg compartment by single-cell RNA-sequencing (scRNA-seq). Once again, to avoid perturbation of their phenotype during the experiment, we isolated Tregs from spleen and bone marrow of *Ikbkb* x Foxp3-GFP+ mice according to GFP expression. We confirmed that mutant Tregs submitted for sequencing followed a similar trend towards increased IL-17 production as previously observed (Fig. S7a, b). UMAP plots of scRNAseq data performed on suspensions from spleen and bone marrow Tregs revealed the nine previously described subsets of Tregs[28] (Fig. 6a, b, Supplementary data 3). We did not identify a unique (previously undescribed) Treg subset but did observe a significant expansion of skin non-lymphoid tissue Tregs (Skin NLT) in both spleen and BM from *Ikbkb*mut mice (Fig. 6b, Supplementary data 4). Hallmarks of Skin NLT include several TNFR superfamily members (*Tnfrsf9, Tnfrsf4, Tnfrsf1b, Tnfrsf8, Tnfrsf18*) as well as conventional Treg genes (*Tigit, Ctla4, Areg*) (Fig. 6c). The Skin NLT subset is enriched for transcription factors related to Th17 differentiation (Fig. 6d). In both compartments, Skin NLTs were increased approximately 2-3 fold, and in the spleen, Skin NLT accounted for approximately 60% of analysed Tregs compared to 30% in wild-type mice (Fig. 6b).

Amongst the top ranked differentially expressed genes in the Skin NLT subset were *Itgae* and *CD69* which mark tissue-resident T cells (Fig. S7c, Supplementary data 5). *Itgae* encodes CD103, an integrin that pairs with integrin β7 to bind E-cadherin, which enables retention in tissues like gut and skin. Indeed, *Itgae* was over expressed in mutant cells relative to WT skin NLTs (Fig. 6c). We re-analysed Tregs for protein expression and also observed a substantial expansion of CD103 + CD4+ T cells in mutant mice (Fig. S8a). There was no significant change in CD103 + CD8+ T cells (Fig. S8a). Furthermore, we observed a substantial increase in CD103+ Tregs in *Ikbkb*mut mice, with evidence of a gene-dose effect (Fig. 6e). Expansion of CD103+ Tregs was evident in spleen, bone marrow and thymus (Fig. S8b). Additionally, to further highlight that the cluster of interest was CD103+, we confirmed that mutant CD103+ Foxp3+ cells produced IL-17, whereas we observed no statistical difference in IL-17 production between WT and mutant CD103- Foxp3+ cells (Fig. 6f, g). Mutant CD103+ Tregs were expanded in previously performed reciprocal bone marrow chimeras (Fig. 5a),

when *Ikbkb* mutation was not confined to non-haematopoietic tissues (Fig. 6h).

The Skin NLT cluster also exhibits increased expression of genes related to tissue (skin) Tregs (Fig. 7a). Previous studies have identified the TNFRSF-NF-κB axis as important for effector Treg formation[49]. Consistent with the expansion of this subset under the influence of *Ikbkb*GoF, analysis of the skin NLT transcriptome against all GSEA hallmark sets identified TNF-signaling via the NF-κB pathway as the most significant association (Fig. 7b, c, Supplementary data 6). In addition, we observed a prominent NF-κB target gene signature within the expanded skin NLT subset (Fig. 7d). Tregs in the skin have also been linked to elevated expression of TGF-β and integrin pathway genes[50]. Consistent with this report, skin NLTs identified in *Ikbkb*GoF mice showed a significant increase of TGF-β pathway genes *Tgfb1, Itgav*, and *Itgb8* compared with other Treg clusters (Fig. S8c). The NLT skin subset of Tregs also revealed significant enrichment of genes associated with TGF-β signalling. Enrichment of genes associated with integrin pathways was not significant (Fig. S8d). Taken together, transcriptome analysis suggests that under normal circumstances, the Skin NLT subset is regulated by NF-κB, and therefore it is driven to expand under the influence of *Ikbkb*GoF. In addition, *Ikbkb*GoF drives acquisition of an IL-17+ effector phenotype that promotes changes consistent with psoriasis or psoriatic arthritis, depending on the magnitude of the defect.

## Discussion

Our principal finding is the identification of a pathway that drives the formation of Foxp3 + CD4+ tissue-resident Tregs to become pro-inflammatory and pathological. This modified Treg population arises spontaneously in vivo as a result of increased activity of the canonical NF-κB pathway, conferred by a GoF missense mutation in *Ikbkb* (p.V203I)[31]. The size of this modified Treg subset varies with the magnitude of IKK2 activity, as both abundance and end-organ consequences of their pro-inflammatory actions are gene-dose dependent. The subset of pathological Tregs identified here exhibits a predilection for skin, bone marrow and joints, where they contribute to inflammatory pathology. The threshold for skin diseases appears to be lower than that for joint and systemic diseases since the former is observed in heterozygous mice, while the latter is only observed in homozygotes. Interestingly, the resultant skin pathology resembles psoriasis, and the systemic disease accurately models psoriatic arthritis, with axial and digital arthritis, characteristic nail changes, and marked inflammatory infiltrates in the bone marrow adjacent to affected joints, mimicking bone marrow oedema.

*Ikbkb*GoF confers the opposite phenotypes to those described for NF-κB loss-of-function (LoF) mutations, which have implicated NF-κB transcription factors c-Rel and p65 in maintaining stable expression of Foxp3. c-Rel deficiency results in a reduction in thymic Tregs, although those that emerge maintain normal suppressor function[11,13,15]. Similarly, human *NFKB1* haploinsufficiency has been shown to result in a selective loss of Helios+ Tregs[49]. Our findings, however, could not have been predicted from these LoF studies, and thus provide further evidence of the power of GoF mutations to yield insight into pathogenic pathways, especially for inflammatory disease. Significantly, we have discovered that IKK2 activity not only regulates Treg abundance, but also specifies expansion of a subset of tissue-resident Tregs, drives them to adopt an effector phenotype, and mediates end-organ pathology.

Normal activation of NF-κB via TNFRSF is important for driving the formation of effector Tregs, and the transcriptome of these cells overlaps with that of Skin NLTs (including *Tigit, Pdcd1, Klrg1, Itgae, Batf*, and *Irf4*)[17,49,51]. Single-cell analysis reveals that the modified Skin NLT subset that forms as a result of *Ikbkb*mut also co-opts element of the Th17 transcriptome, including *Rora, Rorc* and *Stat3*. Surprisingly, the loss of TNFRSF members OX40 and CD27 has also been shown to drive Treg differentiation in the skin[52]. Reduction in Foxp3 expression was

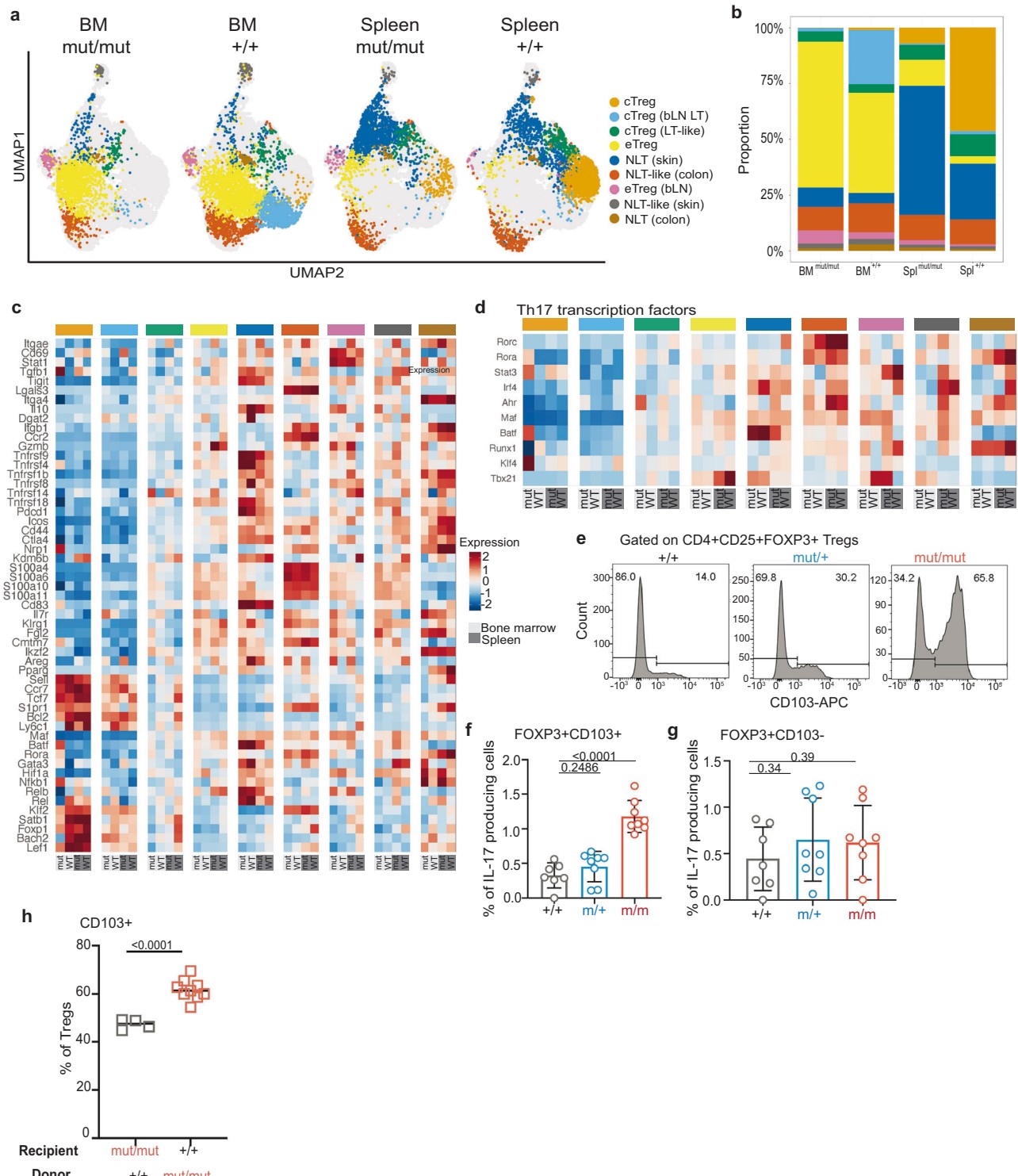

**Fig. 6 | *Ikbkb*^mut Tregs are modified non-lymphoid tissue skin Tregs. a** UMAP plot of scRNASeq analysis of spleen and bone marrow CD4+ Foxp3^GFP Tregs sorted from *Ikbkb*^+/+ and *Ikbkb*^mut/mut mice (*n* = 3 per genotype). Cells are highlighted by cluster. **b** Summary of each cluster as a proportion of all cells within the sample. **c** Average gene expression of established markers for each identified Treg cluster from *Ikbkb*^+/+ and *Ikbkb*^mut/mut mice identified in spleen and bone marrow. Clusters are based on references[28,64]. cTreg = central Treg, eTreg = effector Treg, LT = lymphoid tissue, NLT = non-lymphoid tissue, bLN = brachial lymph nodes. Lower panel: light grey, bone marrow; dark grey, spleen. **d** Average gene expression levels of

established markers from Th17 transcription factor program[65] within each cell population of every Treg sample. Lower panel: light grey, bone marrow; dark grey, spleen. **e** Representative flow cytometric histograms of CD103 expression, gated on CD4^+ CD25^+ Foxp3^+ Tregs from spleen. Summaries of IL-17 producing CD103+ Tregs (CD103+Foxp3+) (**f**) and IL-17 producing CD103- Tregs (CD103-Foxp3+) (**g**) from spleen. *Ikbkb*^+/+, *n* = 7; *Ikbkb*^mut/+, *n* = 8; *Ikbkb*^mut/mut, *n* = 8. **h** Proportion of CD103+ Tregs from chimeras. *Ikbkb*^mut/mut recipients, *n* = 4, *Ikbkb*^+/+ recipients, *n* = 9. Summary graphs show means +/- s.d. One-way ANOVA with Bonferroni's multiple comparison test (**f, g**) and Student's t-test (**h**).

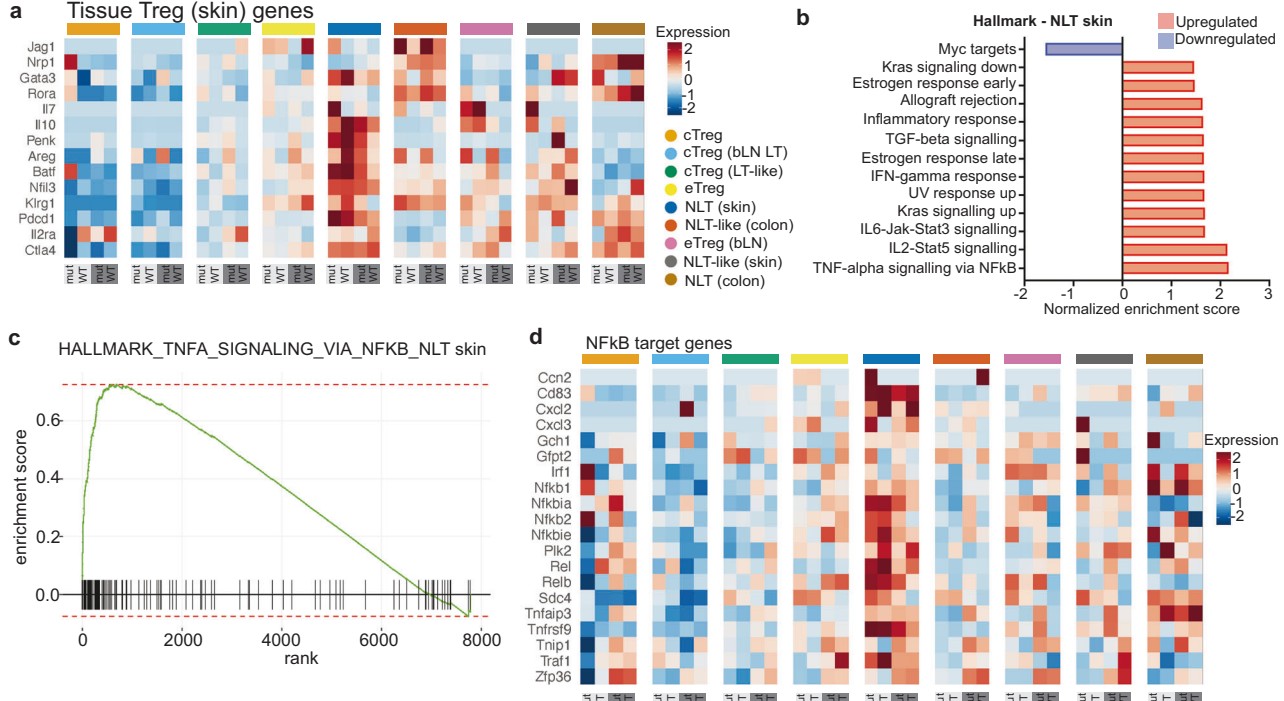

**Fig. 7 | NF-κB transcriptome of non-lymphoid tissue skin Tregs. a** Average gene expression of skin NLT genes[66,67]. Lower panel: light grey, bone marrow; dark grey, spleen. **b** Hallmark pathway analysis showing top 10 ranked significant pathways (defined by normalized enrichment score – NES) in NLT (skin) Treg cluster either up- or down-regulated. **c** Barcode plot showing upregulation of genes in Hallmark pathway TNFA_Signaling_Via_NF-κB in NLT (skin) Treg cluster. NES = 2.1722, *p* adj = 0. *p* values for a collection of gene sets was determined by the fast pre-ranked gene set enrichment analysis (adjusted *p*-value cut-off = 0.05). **d** Average gene expression of NF-κB-dependent genes from reference[68] (GSEA systemic name: M14435). Lower panel: light grey, bone marrow; dark grey, spleen.

shown to result in expression of effector type cytokines in ex-Treg cells[53], and Foxp3 is known to act specifically to suppress IL-17 expression by binding RORγ[54,55]. We saw no change in Foxp3 or CD25 expression in *Ikbkb*[mut] mice, and the majority co-express Helios. The Skin NLT subset also bears hallmarks of tissue-resident Tregs. They exhibit high-level expression of *Itgae* (CD103) and *Cd69*, as well as *Areg, Hif1a, Gzmb, Fgl2, Irf4, Ctla4, Il10* and *Batf*, and downregulation of *Bach2*. Based on the skin NLT transcriptome, and their selective expansion in this model, it appears that NF-κB is a proximal driver of their formation. Our single-cell analysis reveals that even in WT animals, skin NLTs are distinguished by an NF-κB transcriptome, which explains their selective expansion under the influence of enhanced NF-κB activity[28,30,47].

While our principal finding is that overactivity of canonical NF-κB results in the expansion of a pathological subset of Tregs, our model is also remarkable for the pattern of pathology, which by both distribution of inflammation, histopathological analysis and skin transcriptome, exhibits hallmarks of psoriasis in heterozygous mice, and psoriatic arthritis in the homozygous state. At barrier sites such as skin, homeostasis and absence of inflammation appears to depend on a tightly regulated Th17 response, since cutaneous infection is associated with Th17 deficiency and pathology with Th17 excess[56,57]. IL-17-mediated skin diseases include psoriasis, hidradenitis suppurativa (HS) and severe pustular acne[43], and the importance of IL-17 has been confirmed with successful introduction of IL-17 antagonists in the clinic[37,57,58]. Studies of skin biopsies from patients have demonstrated IL-17+ αβ T cells[59], as well as a correlation between the presence of IL-17+ Foxp3+ Tregs in skin and psoriasis[60]. Our findings raise the possibility that these are the same cells, which is supported by recent evidence that Tregs with a Th17-like inflammatory phenotype are present in synovial fluid of patients with spondyloarthritis[61].

Our results provide a plausible pathway to account for the formation of Tregs and development of pathology. We have discovered that increased IKK2 activity drives formation of inflammatory Tregs and have also provided several lines of evidence that the expanded IL-17-producing Tregs contribute to pathology. The model and mechanism of pathology is complex, however, as we show through reciprocal bone marrow chimaera experiments. T cell-intrinsic actions of *Ikbkb*[mut] are necessary for pathology but complete penetrance is likely to depend on the mutation acting simultaneously in non-haematopoietic tissue. This is not surprising since conditional deletion of *Ikbkb* from epithelium has been shown to induce an immune phenotype, and *Card14*[GoF] has been shown to act in the keratinocytes to induce psoriasis and the related disorder, pityriasis rubra pilaris[62]. In our model, a cell-intrinsic *Ikbkb* defect in T cells appears to be necessary for systemic disease. Furthermore, our results indicate that the magnitude of IKK2 GoF can control the transition from skin to systemic disease. This is important as approximately 30% of psoriasis patients develop arthritis in which inflammation is concentrated on the tendon and ligament insertions into bone (enthesitis), digits (dactylitis) and nails.

A causal pathway involving IKK2-mediated NF-κB activation to condition aberrant tissue resident Tregs could explain several features of psoriasis. Tregs residing in human skin contribute to homeostasis after wounds and other cutaneous insults[32,52,54,55,63]. In psoriasis and PsA, keratinocyte proliferation, the Koebner phenomenon, nail changes, and enthesitis all point to exaggerated tissue repair or exuberant wound healing responses. Furthermore, while we have discovered how expansion of Skin NLTs and development of pathology arises as a result of constitutive NF-κB activity conferred by a single mutation, NF-κB is a plausible final common pathway to disease. Thus, the overactive phenotype conferred by *Ikbkb*[GoF] could be replicated by other mutations or environmental stimuli, including factors already implicated in seronegative arthritis. For example, endoplasmic reticulum

stress arising as a result of HLA-B27 misfolding and the intestinal microbiome have been implicated in the aetiology of psoriasis and are both activators of NF-κB. In summary our results provide a molecular and cellular mechanism to explain a wound-healing model of psoriasis and psoriatic arthritis, linked by the magnitude of the NF-κB defect. Further evaluation of this pathway in human disease may be warranted.

## Methods

### Mice

All mice were on the C57BL/6 background. Unless specified, mice were used between 6 weeks and 12 months of age at the time of analysis. When possible, mice were age-matched within experiments, often littermates. A combination of male and female mice were used for experiments. No consistent differences between males and females were observed. All animals were bred and maintained under specific pathogen-free conditions at 18-24 °C and 40–70% humidity with a lighting cycle of 7 am to 7 pm light and 7 pm to 7 am darkness at the Australian Phenomics Facility at the Australian National University (Canberra). Animals were used in agreement with the protocols approved by the ANU Animal Experimentation Ethics Committee. This research operated under ANU Ethics Protocols A2018/06, A2020/21, and A2021/22. To determine genotypes, routing genotyping processes were conducted by the APF Genotyping service. Ear punches from mice were used to extract genomic DNA and sequenced by Sanger sequencing conducted by the Biomolecular Resource Facility (BRF) or Amplifluor assay conducted by the APF Genotyping staff. In each experiment, mice described as $Ikbkb^{+/+}$ were siblings of mice with $Ikbkb^{V203I}$ mutations. The $Ikbkb^{V203I}$ were generated as described previously[31].

### Cell media and buffers

Cell culture media used: Complete RPMI (supplemented with 2% HEPES, 1% penicillin and streptomycin, and 10% fetal bovine serum), complete IMDM (supplemented with 1% penicillin and streptomycin, 10% fetal bovine serum, and 0.1% Beta-mercaptoethanol), and complete DMEM media (supplemented with 1% penicillin and streptomycin, and 10% fetal bovine serum). Flow cytometry wash buffer recipe included 1X PBS supplemented with 2% fetal bovine serum. Red blood cell lysis buffer recipe included $dH_2O$ supplemented with 10% RBX lysis buffer to make final 1% concentration. Skin digestion buffer recipe: RPMI 1640 without phenol red (supplemented with 1 mg/mL collagenase Type IV (Thermo Fisher) and 0.1 mg/mL DNAse-I (Sigma)).

### Antibodies

**Flow cytometry.** The following antibodies were used: 7AAD (Thermo Fisher, cat no. A1310), Rat anti-CD16/CD32 monoclonal antibody unconjugated (BD Biosciences, cat no. 553142, RRID: AB_394657), Brilliant violet 510 anti-mouse CD3 (BioLegend, cat no. 100234, RRID: AB_2562555), Alexa fluor 700 anti-mouse CD19 (BioLegend, cat no. 115528, RRID: AB_493735), Brilliant violet 605 anti-mouse CD45.1 (BioLegend, cat no. 110737, RRID: AB_11204076), Brilliant violet 510 anti-mouse CD45.2 (BioLegend, cat no. 109838, RRID: AB_2650900), APC-Cy7 anti-mouse CD45 (BioLegend, cat no. 103116, RRID: AB_312981), FITC anti-mouse CD4 (BioLegend, cat no. 100510, RRID: AB_312713), PE anti-mouse CD4 (Thermo Fisher, cat no. 12-0042-83, RRID: AB_465511), Alexa fluor 700 anti-mouse CD8a (BioLegend, cat no. 100729, RRID: AB_493702), Brilliant violet 605 anti-mouse CD25 (BioLegend, cat no. 102035, RRID: AB_2563059), PE-Cy7 anti-mouse/human CD44 (BioLegend, cat no. 103029, RRID: AB_830786), APC/Fire 750 anti-mouse CD62L (BioLegend, cat no. 104449, RRID: AB_2629772), APC anti-mouse CD103 (Thermo Fisher, cat no. 17-1031-82, RRID: AB_1106992), Alexa fluor 647 anti-mouse FOXP3 (BioLegend, cat no. 126407, RRID: AB_1089115), PE/Dazzle 594 anti-mouse/human

Helios (BioLegend, cat no. 137231, RRID: 2565797), APC conjugated Rat anti-IFN-gamma (BD Biosciences, cat no. 554413, RRID: AB_398551), FITC anti-mouse IL-17A (BioLegend, cat no. 506908, RRID: AB_536009), Brilliant violet 510 anti-mouse Ly6G (BioLegend, cat no. 127633, RRID: AB_2562937), TCR gamma/delta eFluor 450 (Invitrogen, cat no. 48-5711-82, RRID: AB_2574071, BV421 mouse anti-Ki-67 (BD Biosciences, cat no. 562899, RRID: AB_2686897), Alexa fluor 700 anti-mouse CD45.1 (BioLegend, cat no. 110724, RRID: AB_493733), BUV737 Ms CD45R/B220 (BD Biosciences, cat no. 612838, RRID: AB_2870160), Rat anti-CD19 monoclonal antibody PE (BD Biosciences, cat no. 553786, RRID: AB_395050), PerCP/Cyanine5.5 anti-mouse CD4 (BioLegend, cat no. 100433, RRID: 893330), BUV395 Rat anti-mouse CD8a (BD Biosciences, cat no. 563786, RRID: AB_2732919), Rat anti-CD25 monoclonal antibody (BD Biosciences, cat no. 557192, RRID: 398623), FOXP3 monoclonal antibody FITC (Thermo Fisher, cat no. 11-5773-82, RRID: AB_465243), PerCP-Cyanine5.5 anti-mouse CD3 (BioLegend, cat no. 100218, RRID: AB_1595492), Brilliant violet 605 anti-mouse CD4 (BD Biosciences, cat no. 563151, RRID: AB_2687549), Pacific blue anti-mouse/human CD44 (BioLegend, cat no. 103020, RRID: AB_493683), Brilliant violet 711 anti-mouse CD25 (BioLegend, cat no. 102049, RRID: AB_2564130), PE anti-mouse CD103 (BioLegend, cat no. 121405, RRID: AB_535948), IL-17A monoclonal antibody APC (Thermo Fisher, cat no. 17-7177-81, RRID: AB_763580), IFN gamma monoclonal antibody PE-Cyanine7 (Thermo Fisher, cat no. 25-7311-82, RRID: AB_469680), Rat anti-CD4 monoclonal antibody Alexa fluor 700 (BD Biosciences, cat no. 557956, RRID: AB_396956), Ms CD45.1 BUV737 (BD Biosciences, cat no. 612811, RRID: AB_2870136), Rat anti-CD19 APC (BD Biosciences, cat no. 550992, RRID: AB_398483), Brilliant violet 605 anti-mouse CD62L (BioLegend, cat no. 104438, RRID: AB_2563058).

### Lymphoid tissue preparation

Mouse lymphoid organs (spleen and lymph nodes) were collected in cold RPMI-640 (Thermo Fisher) with 10% FBS (ThermoFisher). Spleen and lymph nodes were disrupted and processed into single cell suspension by syringe plunger and passaged through 70 µm filters (Miltenyi Biotec), followed by red blood cell lysis process (BD).

For bone marrow harvest, the tibia and femur were cleaned and cut at both ends to allow for bone marrow to be flushed out. These bones were collected into empty 200 µL microcentrifuge tubes; these tubes had a small hole at the bottom tip which was created by pushing a 16-gauge needle through. These 200 µL tubes were then placed into 1.5 mL microcentrifuge tubes with the lid removed. Samples were centrifuged at high speed for 10 seconds which enabled bone marrow to be flushed out to the bottom of the 1.5 mL tubes. Cell pellets were then resuspended in culture media and passed through a 70 µm cell strainer to prevent cell clumping.

### Blood processing from retro-orbital bleed

200µL blood from the mouse via a retro-orbital bleed was extracted and placed into heparinised capillary tubes for transfer into cluster tubes containing 20 µl of EDTA. Blood was then incubated with 1x Pharmlyse lysing buffer (BD) diluted with $dH_2O$ for 5 minutes at room temperature. Blood samples were then spun down at 400 x $g$ for 5 minutes at room temperature. The supernatant was removed and lysis step was repeated twice more. Cells were then resuspended in FACS wash buffer and kept on ice until required for downstream applications (e.g. flow cytometry).

### Skin tissue preparation

Mice were culled by cervical dislocation and shaved from behind the ears to the base of the tail, and down both flanks using an electric veterinary shaver. Depilatory cream (Veet) was applied for 3-5 minutes, then scraped off using a plastic spatula, and mice were washed thoroughly under running water and then dried with a paper towel. Ears

and tails were removed and the tail skin dissected. A large patch of hairless skin approximately 5 cm x 5 cm was dissected from the back/flank, with care taken to clean the underside of fat and lymph nodes. For each animal, the three skin samples (ears, tail, and back/flank) were treated individually as follows.

Tissues were transferred to 60 mm petri dishes, minced very finely using dissecting scissors, and incubated in 6 mL RPMI 1640 without phenol red (Gibco) supplemented with 1 mg/mL collagenase Type IV (Thermo Fisher) and 0.1 mg/mL DNAse-I (Sigma) for 60 minutes in a humidified 37 °C/5% $CO_2$ incubator, with gentle agitation every 15 minutes.

After incubation, digested tissue fragments were transferred to 70 μm cell strainers (Corning), mashed through using 3 mL syringe plungers (Terumo) and collected in 50 mL tubes. Periodically, the strainers were washed with PBS (Gibco)/5 mM EDTA (Sigma) to a total volume of 15 mL per sample. The suspensions were centrifuged at 300 x $g$/5 minutes/4 °C. At this point, debris was removed from the suspensions using Debris Removal Solution (Miltenyi Biotech). Briefly, pellets were resuspended in 3.1 mL ice cold PBS, 0.9 mL debris removal solution was added, solutions were mixed slowly 10 times using a 5 mL serological pipette and layered beneath 4 mL ice cold PBS in a 15 mL tube. Tubes were centrifuged at 3000 x $g$/10 minutes/4 °C with 50% acceleration and brake. The supernatant was aspirated, pellets were resuspended in 3 mL ice-cold PBS, filtered through a 40 μm filter (Corning), centrifuged at 500 x $g$ for 5 minutes, resuspended in 0.2 mL PBS and counted.

### Flow cytometry

Processed single-cell suspensions were transferred to 96-well round-bottom plates. Plate was centrifuged at 400 x $g$ for 5 minutes at 4 °C, and the supernatant flicked off. A volume of 50 μL of antibody cocktail mixtures were added to each sample pellet and incubated for 20 minutes on ice. If required, cells were fixed and permeabilised using the Fixation/Permeabilization kit (eBiosciences; cat# 00-5523-00), as described by the manufacturer. Cells then underwent intracellular staining with an appropriate antibody cocktail for 30 minutes on ice. Samples were acquired on an LSR II or LSR Fortessa flow cytometer (BD). Automatic fluorochrome compensation was used and manually improved after acquisition. FlowJo software V10 (Treestar) was used for analysis.

### Flow cytometric cell sorting

Harvested spleen and lymph nodes were processed as described above. Cells were briefly pre-treated with 1 x red blood cell lysis buffer for 15–30 seconds at room temperature and centrifuged with 1x PBS at 400 x $g$ for 5 minutes at 4 °C. Supernatant was removed and cell pellet was then surface stained with antibody cocktail and the processed as described above. Cells were resuspended in FACS wash buffer prior to cell sorting on a FACS Aria II or FACS Fusion (BD) and was performed by experienced staff of the JCSMR Imaging and Cytometry facility, under sterile conditions.

### Ex vivo stimulation with PMA/Ionomycin for cytokine production

Cells were plated onto a 96-well round-bottom plate and centrifuged at 400 $g$ for 5 minutes at 4 °C and supernatant flicked off. Cells were then stained with primary antibody cocktail as described above. After primary stain, cells were centrifuged to the same conditions and flicked off. Cells were then stimulated for 6 hours in complete IMDM with 50 ng/mL PMA, 100 ng/mL Ionomycin with GolgiStop (BD Biosciences) added after 2 hours of stimulation. During stimulation, cells were incubated at 37 °C. Stimulation was halted by adding cold 1x PBS to samples, then centrifuged and supernatant flicked off. Cells were stained with live/dead stain for 30 minutes on ice. After centrifuging

and flicking, cells were fixed and permeabilized using the fixation kit described above. Intracellular staining with cytokine mix was then performed. Cells were incubated with cytokine mix for 30 minutes on ice.

### T cell polarization

Naïve T cells (CD4+CD62L$^{hi}$CD44$^{lo}$) were FACS-sorted as described above. For Th17 polarization, $4 \times 10^5$ cells were cultured in 48-well flat-bottom plates with complete IMDM with 3 μg/mL plate-bound anti-CD3ε (BD, 553058), 2μg/mL anti-CD28 (BD, 553295), 2.5 μg/mL anti-IL-4 (clone: 11B11, BioXCell, BE0045), 2.5 μg/mL anti-IFNγ (clone: XMG1.2, BioXCell, BE0055), 0.5 ng/mL TGFβ (Miltenyi Biotec, 130-095-067), and 30 ng/mL IL-6 (Miltenyi Biotec, 130-094-065). After 72 hours of incubation at 37 °C, cells were split and incubated in fresh media and cultured for a further 24 hours. On day 4, cells were harvested and stained for surface markers for 20 minutes according to above. Cells were then stimulated for 6 hours in complete IMDM with 50 ng/mL PMA, 100 ng/mL ionomycin with GolgiStop added for 2 hours of stimulation. During stimulation, cells were incubated at 37 °C. Fixation, permeabilization and intracellular staining was performed as described above.

FACS-sorted naïve T cells were cultured with 3μg/mL plate-bound anti-CD3ε (BD), 2μg/mL anti-CD28 (BD), 2.5 μg/mL anti-IL-4, 0.5 μg/mL mIL-2 (Peprotech, 212-12), and 0.4 μg/mL IL-12 (Miltenyi Biotec, 130-096-707). After 72 hours of incubation at 37 °C, cells were harvested, surface stained and re-stimulated with PMA/Ionomycin and GolgiStop for 6 hours in the same conditions described above. Fixation, permeabilization, and intracellular staining were then performed as described above.

### In vitro Treg suppression assay

$2.0 \times 10^4$ naive T cells (CD4$^+$CD62L$^{hi}$CD44$^{lo}$CD25$^-$) were FACS-sorted from donor mice and used as'responders'. These cells were CTV labelled as described below. Responders were co-cultured with FACS-sorted Tregs (CD4$^+$CD25$^+$Foxp3-GFP+) at ratios of 0:1, 1:1, 3:1 (Tregs:responders), along with $4.0 \times 10^4$ APCs (CD3$^-$CD4$^-$CD8$^-$). Cells were stimulated in the presence of 3 mg/mL plate-bound anti-CD3ε (BD Biosciences) for 3 days at 37 °C. On day 3, cells were stained with appropriate antibodies and analysed by flow cytometry.

### Cell trace violet (CTV) labelling

Cells were washed with 1x PBS. Cells were stained with prepared CTV reagent (Thermo Fisher, C34557) at a concentration of 1-15 $\times$ 10$^6$ cells per mL. Cells were incubated in the dark at room temperature for 20 minutes then washed with cold PBS before use.

### Single cell RNA-sequencing

Treg (CD4$^+$CD8$^-$FOXP3-GFP+) cell population was isolated from spleen tissue and bone marrow by FACS-sorting from 3 mice of each genotype (WT and homozygous). Sorted cells were then counted using a haemocytometer. Cells were washed with FACS wash and centrifuged at 400 $g$ for 5 minutes at 4 °C. Washing and centrifugation steps were repeated twice more. After wash steps, samples of the same cell type and genotype were pooled, and each pooled sample was counted using Countess® Automated Cell Counter (Thermo Fisher) to determine concentration.

A droplet-based scRNA-seq technique was conducted using a Chromium system (10X Genomics). Cell counts were used to enable capture of approximately 10,000 cells. The protocol for the Chromium NextGen Single Cell 5' Reagent kit (Dual index) was followed according to manufacturer's instructions. Samples were sequenced on an Illumina NovaSeq 6000 according to manufacturer's instructions. Sequencing data was aligned and quantified using the CellRanger (v6) software package.

## Single-cell RNA-sequencing analysis

CellRanger processed data were analyzed with Seurat (v4) using R in RStudio. As detailed by the Satjia lab (Hao et al., 2021; Satjia-Lab., 2021), a standard unsupervised clustering workflow including QC and data filtration, calculation of highly variance genes, dimension reduction, clustering and the identification of cluster markers was performed for each of the samples. An integrated dataset for the sample was generated by integrating Treg samples from bone marrow and spleen, with a total of 4 samples (2x WT, 2x homozygous/mutants).

To generate an overview of the datasets showing the relationships between cell population clusters, principal component analysis (PCA) (dims = 30) was performed and visualised by UMAP. The integrated data (Treg) was normalized using the SCTransform function. PCA and UMAP were calculated using the RunPCA and Run UMAP functions, respectively. Functions used in the analysis included FindAllMarkers, FindConservedMarkers, and FindMarkers.

To identify clusters in the data, we used FindNeighbours then FindClusters function from Seurat, with the same number of principal components (dims = 30) used for clustering and visualised in UMAP. Cluster annotation was done by inspecting markers detected by the FindConservedMarkers function. *Trbv* family genes were removed to eliminate cluster domination dictated by differences in expression of these genes. Global clustering of the integrated data (Treg) was done with the resolution parameter set to 0.3 with the Louvain algorithm. After clustering the integrated dataset (Treg), we excluded a B cell population characterised by high expression of B cell-related genes (e.g. *Cd19, Cd79a*). An additional population that lacked expression of T cell-related genes (e.g. *Cd3e, Cd4*) was also excluded. Lastly, to generate expression heatmaps, the AverageExpression function was used to calculate average expression of the genes within each cluster. Gene set enrichment analysis was performed using the fgsea package (V.1.23.4). Gene expressed by more than 5% of cells were ranked by log fold-change of average expression, the gene set enrichment analysis tests were performed based on the Molecular Signatures Database (MSigDB) with the threshold of adjusted $p$ value < 0.05.

## Bulk RNA-sequencing sample preparation

Skin from ears and tail of each mouse ($n = 3$ per genotype) was taken and snap froze in liquid nitrogen. Samples were defrosted in RNA protect reagent, and mechanically cut into smaller pieces of 3–4 mm. We used Monarch total RNA extraction kit to isolate total RNA from the samples. Briefly: 300 μl of RNA protection was added to each sample followed by 45 μl of Prot K Reaction buffer plus Prot K and incubated at 55 °C for 30 minutes. Samples were vortexed and spun at 16,000 x $g$ for 2 minutes and supernatant transferred to fresh microfuge tubes. An equal volume of RNA Lysis buffer was added and samples were transferred to a gDNA removal column, spun at 16000 x $g$ for 1 minute. An equal volume of 95% ethanol was added to the flow through and transferred to a purification column and spun at 16,000 x $g$ for 1 minute. Flow through was discarded. 500 μl RNA wash buffer was added to the column and spun for 30 seconds, then 80 μl of DNase plus DNase I reaction buffer was added and incubated for 15 minutes at room temperature. 500 μl RNA priming buffer was added and spun for 30 seconds, flow through was discarded, and the column was washed twice with 500 μl of wash buffer. 40 μl nuclease-free water was added to the column and spun for 30 seconds to collect the total RNA in fresh nuclease-free microfuge tubes.

## Bulk RNA-sequencing analysis

All samples were evaluated to be of good quality by FastQC. Reads were aligned to the mouse genome GRCm38 (mm10) assembly by HISAT2 (v2.1.0) and sorted with SAMtools. Mapped reads were assigned and counted based on NCBI Refseq gene annotations by FeatureCount (v1.4).

After filtering out genes expressed at low levels across all the samples, the remaining genes with at least 1 count per million (CPM) reads in more than three samples were included for downstream analyses. Gene read libraries were normalised by the trimmed mean of M values (TMM) method and data were transformed by voom with sample quality weight adjustment. The data were fitted in a linear model with moderated t-statistics with the empirical Bayes method, and differentially expressed genes with a Benjamini–Hochberg adjusted $p$-value < 0.05 were identified and visualised with Glimma and IGV. GSEA was performed by the CAMERA method accounting for intergene correlation based on the MSigDB. All statistical analyses were performed by the limma package.

## Haematology

Blood samples were obtained from mice via retro-orbital bleed using heparinised capillary tubes and transferred into cluster tubes containing 20 μL of EDTA. 200 μL of whole blood was loaded onto the Advia 2120 Haematology Analyser (Siemens), with results exported into a Microsoft excel document.

## Histopathology

For specific post-mortem tissue analysis, tissues were collected and fixed in a 10% neutral buffered formalin solution for a minimum of 24 hours. After fixation, samples were submitted to either the Pathology department at The Canberra Hospital or the Australian Phenomics Network (APN) Histology and Organ Pathology at the University of Melbourne, where they were embedded in paraffin, sectioned onto scope slides and stained with hematoxylin and eosin (H&E). Stained slides were examined by microscopy using an Axio Scan.Z1 microscope (Zeiss).

Live mice were submitted to the Australian Phenomics Network Histopathology and Organ Pathology Service, University of Melbourne, Australia, for comprehensive pre- and post-mortem examination.

## Serum cytokine analysis

Serum cytokines were measured by Mesoscale according to the manufacturer's instructions using Mouse Isotyping Panel 1 kit (K15183B; IgA, IgG1, IgG2a, IgG2b, IgG3, IgM) and Mouse U-Plex Biomarker Group 1 (K15069L-2; IFNα, IFNβ, IFNγ, IL-1β, IL-2, IL-6, IL-10, IL-12p70, IL-17A, TNFα, IL-4, IL-5). For Isotyping kit K15183B, plasma was diluted 1: 100,000 in 1% FBS/ PBS. Washes were performed in 0.05% PBS-tween (Merck, P7949). Plates were read by the MESO SCALE Discovery Sector Imager 6000 and analysed on R-Studio software. The lower limits of detection (LLOD) per cytokine were as follows: IFNα, 140; IFNβ, 5.2; IFNγ, 0.16; IgA, 33; IgG1, 45; IgG2a, 24; IgG2b, 11; IgG3, 34; IgM, 22; IL-10, 3.8; IL-12p70, 48; IL-17A, 0.3; IL-1B, 3.1; IL-2, 1.1; IL-33, 2.2; IL-4, 0.56; IL-5, 0.63: IL-6, 4.8; TNF, 1.3 (all units in pg/mL).

## Mixed bone marrow chimeras

*Rag1*-deficient recipient mice were sub-lethally irradiated with a single dose of 5 Grays. Each recipient mouse was injected with $1 \times 10^6$ bone marrow cells harvested from femurs and tibias of CD45.1 B6 donor mice, combined with $1 \times 10^6$ bone marrow cells harvested from femurs and tibias of *Ikbkb*$^{V203I}$ mice. Cells were re-suspended in 1X PBS for injections. Recipient mice were maintained on antibiotics for 6 weeks, then analysed at 12 weeks post-irradiation.

## 100% bone marrow chimeras

Homozygous *Ikbkb*$^{V203I}$ and WT mice were irradiated and reconstituted with $2 \times 10^6$ bone marrow cells harvested from femurs and tibia given by I.V. injection.

## Adoptive transfers

For in vivo Treg suppression assays (Fig. 4), naïve T cells (CD4+CD62L^hiCD44^lo) and Tregs (CD4+CD25+Foxp3-GFP+) were FACS-sorted from spleens and lymph nodes of *Ikbkb* mice. $4.0 \times 10^5$ naïve T cells and $2.0 \times 10^5$ Tregs were resuspended in 200 µL of 1x sterile PBS and injected intravenously into the tails of recipient mice. Recipients were monitored for health and body mass for 6 weeks or sacrificed earlier if body mass declined by >15%, when organs were collected for histology.

For transfer of *FOXP3*^GFP Tregs into *Rag1*^−/− x *Ikbkb*^V203I mice (Fig. 5), $3.5 \times 10^5$ Tregs (CD4+CD25+Foxp3-GFP+) were FACS-sorted from spleens and lymph nodes of *Ikbkb* mice, resuspended in 200 µL of 1x sterile PBS and injected intravenously into the tails of recipient mice.

## Statistical analysis

Two-tailed unpaired *t*-tests, one-way ANOVA and two-way ANOVA with Bonferroni's multiple comparison test were performed according to the experimental designs. Unless specified otherwise, summary graphs show data from 2-3 independent experiments. Dermatitis and limb inflammation incidence data were compared by Log-Rank (Mantel-cox) test. Statistical analysis was performed using either Rstudio or GraphPad Prism (8.0.1).

## Reporting summary

Further information on research design is available in the Nature Portfolio Reporting Summary linked to this article.

## Data availability

Single cell RNA-sequencing data and bulk RNA-sequencing data were deposited to the NCBI and accessible under the GEO accession numbers: GSE228385, and GSE248917. Source data are provided with this paper.

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

## Acknowledgements

The authors thank the Australian Phenomics Facility staff for husbandry and genotyping, and the Flow Cytometry facility, Biomolecular Resource Facility, the Phenomics Translation Initiative team, and the ANU Bioinformatics Consultancy at the John Curtin School of Medical Research for their services. This work was funded by the National Health and Medical Research Council Program Grant APP1113577 (CGV, MCC), CRE APP1079648 (CGV, MCC) and Project Grant APP1107464 (MCC); the Alan Harvey CVID Research Endowment (CC); Royal Society Wolfson Fellowship RSWF\R2\222004 (MCC). This study utilized the Australian Phenomics Network Histopathology and Organ Pathology Service of the University of Melbourne. The Phenomics Translation Initiative is supported by the Medical Research Future Fund (EPCD000035).

## Author contributions

C.C. designed and performed experiments, analysed the data, and wrote the first draft of the manuscript. Y.H., K.K., A.R.D., M.B.D., N.A.R., J.D.P., R.A.H., J.L. and R.C. performed experiments. C.C., Y.H. and Z.-P.F. analysed gene expression data. A.E. and C.G.V. provided intellectual contributions to experimental design and analysis. B.M. assisted with data analysis and experimental designs and project supervision. All authors contributed to drafting the manuscript. M.C.C. conceived the project, supervised experiments and data analysis and wrote the manuscript.

## Competing interests

The Authors declare no competing interests.
