## [Peer Review File · Nature Communications]

IKK2 controls the inflammatory potential of tissue-resident regulatory T cells in a murine gain of function modelREVIEWER COMMENTS

Reviewer #1 (Remarks to the Author):

In the manuscript by Cardinez et al 2023, the authors investigated the consequences of a mutant IKK2 overexpression, resulting in the overactivation of IKK2. The study is based on an initial study published by the same group a couple of years before, in which they found that IKK Val203Ile mutation (gain of function GoF) in mice and humans results in an immune deficiency syndrome (Cardinez et al. 2018, J Ex Med). In the new manuscript, the authors extended their analysis looking more closely at the consequences of heterozygous and homozygous IKK2 GoF expression in mice.

They found that heterozygous overexpression triggers a psoriasis-like dermatitis in mice by age whereas homozygous overexpression induces psoriasis-like dermatitis and arthritis. Subsequent analysis revealed expansion of IL-17A-expressing Tregs in those mice, which could trigger the disease by injecting them in IKK2 overexpressing and Rag ^{-/-} mice. Overall, the data looks convincing and conclusive. The data is new and relevant for the field. However, some key experiments are missing to get an experimental proof of the claims made in the manuscript. Therefore, I recommend revising the manuscript.

In general: Do patients with IKK2 Val203Ile mutation get psoriasis or show signs of psoriasis arthritis? This would be valuable data to add.

Figure 1.

The authors show that mice develop a dermatitis-like phenotype at the tails and ears. Moreover, they describe a “patchy” inflammation of the skin, however photos of the back skin and a detailed analysis of the phenotype are missing. Especially, the authors state that the skin inflammation resembles psoriasis due to the prevalence of munro abscesses, which are barely shown. The authors should clearly investigate the phenotype in more detail, by flow cytometry analysis of T-cells, neutrophils, etc, as well as by a detailed analysis of psoriasis markers (protein levels of IL-17A, TNF, IL-23, IL-36G or qPCR markers of elevated chemokines, AMPs, and cytokines) to confirm that the observed skin phenotype resembles psoriasis. Moreover, markers of atopic dermatitis, for example, should be investigated, to discriminate their phenotype from other skin pathologies.

In Fig. S1B-C the authors investigated leukocytes from mutant mice. Does this data derive from tails, ears, or back skin? All analyses should be made in both tail/ears and back skin, as skin composition and homeostasis differ between these different sites.

Figure 2

The authors describe that homozygous IKK2 mutant mice show signs of systemic inflammation leading to joint inflammation and arthritis. In further experiments a detailed inflammation of the bones is described, however, analyses of the blood or other organs are missing. Do the authors find an increased circulation of IL-17A-expressing Tregs elsewhere? Do IL-17A-expressing Tregs only localize to the joints and skin or somewhere else?

Figure 3

The authors claim that IKK2 mutant Tregs start to express IL-17A, however, an experimental proof is missing that it is not the other way around, that effector T-cells/Th17 cells start to express FoxP3 thereby turning into IL-17A-expressing FoxP3⁺ T-cells. It would be interesting to see, if an inducible overexpression of mutant IKK2 can convert Th17 cells to Tregs or the other way around that inducible overexpression of mutant IKK2 can induce IL-17A expression in Tregs. This is of special relevance, as normally FoxP3 is known to repress IL-17A expression, and the authors show that FOXP3 Expression is not affected in those cells.

Figure 5

The authors labeled Foxp3-positive Tregs with GFP and injected those cells into IKK2mut or IKK2 wt mice on a Rag1 -/- background. While both mice show no signs of psoriasis-like dermatitis (at least until 21 weeks of age), injection of mutant IKK2 Tregs into mutant IKK2 mice induces the previously described phenotype. These results are important, however, they also imply several more open questions which should be addressed:

- As IKK2 mutant Tregs can only induce the disease in IKK2 mutant animals, it seems that IKK2 overactivation in another cell type, such as keratinocytes, is important for disease onset. The authors should give some experimental insights into this. Does IKK2 mutation activate a keratinocyte or fibroblast response, that induces skin homing of IL17A-expression Tregs? This could be, at least, addressed in vitro.

- Where do IKK2mutant Tregs localize upon injection? Do they home to the skin? The authors could easily test this by detecting the cells via GFP. If the cells do not home to the skin or joints, how could they induce psoriasis and arthritis then? If they are homing to the sites of inflammation, how? Which ligands-receptors might be involved? The results of such investigations should give more mechanistic insights how these Tregs can specifically induce skin and joint inflammation.

Related to Figure 6/7:

The authors claim that mutant IKK2 leads to overactive NF-kB thereby inducing IL-17A expression. This could be easily tested by applying an NFkB inhibitor in vitro on these Tregs and testing, if IL17A expression depends on NF-kB in these cells or not. Especially as the authors also show that RORC and STAT3 seem to be regulated by IKK2. Why does mutant IKK2 only affect IL-17A expression but not TNF or IFNG? The authors should add some data on the IKK2-dependent regulation of IL17A or at least discuss this point.

In general:

The method section lacks some fundamental information. It is not specified if the results are similar in female and male mice. The origin of antibodies is not described. Information on digestion enzymes, etc is missing.

Reviewer #2 (Remarks to the Author):

In their study Cardinez et al. dissect the phenotype resulting in mice with an *Ikkkb* gain-of-function mutation. Their analysis reveals dose dependent (mut vs het) inflammatory disease characteristic of psoriatic arthritis. Skin pathology is especially evident and occurs even in a proportion of het animals. Analysis of lesional and circulating immune populations indicates preferential expansion of Tregs, especially those with a tissue resident phenotypic and transcriptional signature. Surprisingly these Tregs also exhibit elevated IL-17 expression. The authors provide convincing evidence that it is these IL-17 producing Tregs that contributes to pathology. Overall, this is a robust study of broad interest that defines a potentially important role for IKK2 in Treg function. However, the mechanisms by which *Ikkkb* mutation produces this Treg activity are not clearly defined. I also have specific concerns about the single cell RNA-seq reference population used for analysis of skin

resident Tregs. These concerns and suggestions for improvement are described below:

1. The concept of Tregs contributing to inflammatory processes is not new and needs to be more fully discussed. Line 97 they cite an example of conversion but there are other contexts that have been described, especially for tissue resident Treg. ie cite and discuss: DOI: 10.1126/science.1155209; DOI: 10.1126/sciimmunol.abg2329
2. Is there any evidence of inflammation/pathology in trunk skin? Due to higher hair follicle density, there are typically more Tregs in these sites so if not discussion would be warranted.
3. Given that pathology likely hinges on the absolute quantity of IL-17 produced by Tregs their cell number (particularly for Fig. 1K/skin; ie cells/g or cells/cm²) should be presented in addition to frequency. Also in Fig. 1K the representative flow plot does not show convincing Foxp3+ staining. The authors could perform more robust quantification with their Foxp3-GFP reporter cross.
4. The mixed bone marrow chimera in Figure 3D-F is a compelling experiment but only showing 3 wt/het and 2 wt/hom mice is not sufficient to be convincing. The expansion is less than 2 fold in the hom so strong statistical analysis would be helpful.
5. In Fig 4A the authors state there is no significant difference in their suppression assay. It actually looks like the mut/mut has better suppression. Please quantify to make this assertion.
6. The authors have demonstrated that the *Ikbkb* gof mutation leads to Treg expansion and simultaneous preferential expression of IL-17 but I'm unclear if and how these two Treg behavioral outcomes are linked. Is there evidence of enhanced survival or proliferation of the CD103+ Tregs with the *Ikbkb* mutation? Is IL-17 expression elevated compared to CD103+ wild type skin Tregs? Also does the *Ikbkb* gof lead to increased IL-17 production/secretion on a per cell basis?
7. scRNA-seq is performed on splenic and bone marrow Tregs from mutant mice and analysis and compared to the dataset in reference 28. From this it is concluded there is an expansion of Skin NLT. GSEA is used to identify enrichment of TNF-signaling/NF-kB pathway in these Skin NLT. While ref 28 does a good job broadly identifying Treg cell states this data set actually only contains 154 skin derived Tregs. This is too small/biased to make reliable conclusions for the types of analyses performed here. Expand analyses with skin specific datasets. For example: DOI: 10.1126/sciimmunol.abg2329.

Point-by-point response to reviewers

Reviewer 1

In the manuscript by Cardinez et al 2023, the authors investigated the consequences of a mutant IKK2 overexpression, resulting in the overactivation of IKK2. The study is based on an initial study published by the same group a couple of years before, in which they found that IKK Val203Ile mutation (gain of function GoF) in mice and humans results in an immune deficiency syndrome (Cardinez et al. 2018, J Ex Med). In the new manuscript, the authors extended their analysis looking more closely at the consequences of heterozygous and homozygous IKK2 GoF expression in mice.

They found that heterozygous overexpression triggers a psoriasis-like dermatitis in mice by age whereas homozygous overexpression induces psoriasis-like dermatitis and arthritis. Subsequent analysis revealed expansion of IL-17A-expressing Tregs in those mice, which could trigger the disease by injecting them in IKK2 overexpressing and Rag -/- mice. Overall, the data looks convincing and conclusive. The data is new and relevant for the field. However, some key experiments are missing to get an experimental proof of the claims made in the manuscript. Therefore, I recommend revising the manuscript.

In general: Do patients with IKK2 Val203Ile mutation get psoriasis or show signs of psoriasis arthritis? This would be valuable data to add.

This is an interesting question. It is not possible to draw firm conclusions because the syndrome is rare with very few patients reported to date. In the patients described, however, manifestations have included acne conglobata and hidradenitis suppurativa, both of which are in the spectrum of Th17-mediated disorders that includes psoriasis, and which occur with increased frequency than expected by chance in patients with psoriasis [1,2].

Figure 1.

The authors show that mice develop a dermatitis-like phenotype at the tails and ears. Moreover, they describe a “patchy” inflammation of the skin, however photos of the back skin and a detailed analysis of the phenotype are missing. Especially, the authors state that the skin inflammation resembles psoriasis due to the prevalence of munro abscesses, which are barely shown. The authors should clearly investigate the phenotype in more detail, by flow cytometry analysis of T-cells, neutrophils, etc, as well as by a detailed analysis of psoriasis markers (protein levels of IL-17A, TNF, IL-23, IL-36G or qPCR markers of elevated chemokines, AMPs, and cytokines) to confirm that the observed skin phenotype resembles psoriasis. Moreover, markers of atopic dermatitis, for example, should be investigated, to discriminate their phenotype from other skin pathologies.

We thank the reviewer for this question and acknowledge that in our original manuscript, we relied on the histological evidence to suggest dermatitis consistent with psoriasis. Noteworthy changes include hyperkeratosis, parakeratosis, and acanthosis of the epidermis, patchy neutrophil accumulations in the epidermis, and a lymphocytic infiltrate in the dermis. Of course, in homozygous mice, these skin changes occur in the context of systemic disease characterised by spondylitis, destructive changes of the digits consistent with arthritis mutilans, and nail changes. Collectively, these changes suggest *Ikkkb*^{GoF} represents a high-fidelity model of psoriasis and psoriatic arthritis.

In order to respond to the important question raised by the reviewer, we performed an unbiased survey of transcription by performing RNAseq on samples obtained from involved skin (ears and tail) and healthy controls. We analysed the transcriptomes according to the genes mentioned by the reviewer, and also according to a head-to-head comparison of human psoriasis and atopic dermatitis (AD) based on transcriptomes of cells recovered from lesional skin by tape-stripping [3]. We observed significant up regulation of *Tnf*, *Il23*, *Il36g*, as well as other key markers of psoriasis (*Ccl3*, *S100a/b* molecules), whereas genes characteristic of AD are not significantly upregulated. Overall, there is a striking

concordance between ears and tails and more remarkably, between genes identified in human psoriasis and those detected in the *Ikkkb^{GoF}* mouse model. These results provide strong support for the contention that the mouse models psoriasis.

These results are shown below (**Rebuttal Figure 1**) and have been added as **Fig. 1K** of the revised manuscript.

Rebuttal Figure 1. Volcano plots summarising results of RNAseq performed on whole skin from ears and tails of affected *Ikkkb^{mut/mut}* and *Ikkkb^{WT}* mice. Genes implicated in psoriasis (*S100a8*, *S100a9*, *Defb4*, *Tnf*, *Il36a*, *Il36g*, *Nos2*, *Ccl20*, *Ccl2*, *Il1B*, *Cxcl9*, *Il23a*) are labelled. Th2-related genes that have been noted in AD are also shown as they are either downregulated (*Il34*) or not significantly different (*Ccl24*, *Ccr4*, *Ccl17*).

In Fig. S1B-C the authors investigated leukocytes from mutant mice. Does this data derive from tails, ears, or back skin? All analyses should be made in both tail/ears and back skin, as skin composition and homeostasis differ between these different sites.

Skin changes are observed in the back, although later than those in tail and ears. Furthermore, our ascertainment of back involvement is less complete than for ear and tail involvement because it often only becomes apparent after shaving the mice and this procedure is not a routine component of our assessment of extant mice under our current ethics protocol.

In response to the suggestion by the reviewer, we have now analysed involved back skin and confirmed that this is an enrichment for Tregs, similar to that observed in tail and ears. We observed a significant expansion of CD25⁺ Foxp3⁺ Tregs as a proportion of CD4⁺ T cells in the back and tail skin of heterozygous and homozygous mice relative to WT. These results are shown below (**Rebuttal Figure 2**) and have been included in the revised manuscript as **Fig. 1L-N**.

Rebuttal Figure 2. Analysis of CD4⁺ T cells recovered from ears, tail and back skin.

Figure 2

The authors describe that homozygous *IKK2* mutant mice show signs of systemic inflammation leading to joint inflammation and arthritis. In further experiments a detailed inflammation of the bones is described, however, analyses of the blood or other organs are missing. Do the authors find an increased circulation of IL-17A-expression Tregs elsewhere? Do IL-17A-expressing Tregs only localize to the joints and skin or somewhere else?

An unbiased assessment of the pattern of end-organ inflammation was determined by necropsies performed by pathologist blinded to mouse genotypes. All organ systems were examined macroscopically and by histology. Once we obtained these results, we recognised that the pattern of inflammation and tissue damage resembled psoriasis and psoriatic arthritis. Next, we characterised the inflammatory infiltrates and lymphoid organs. We show that Foxp3⁺ cells are expanded in pooled lymph nodes, spleen, bone marrow and thymus (Supp Fig. 3), with a proportionate increase in IL-17A⁺ cells.

Figure 3

The authors claim that *IKK2* mutant Tregs start to express IL-17A, however, an experimental proof is missing that it is not the other way around, that effector T-cells/Th17 cells start to express FoxP3 thereby turning into IL-17A-expressing FoxP3⁺ T-cells. It would be interesting to see, if an inducible overexpression of mutant *IKK2* can convert Th17 cells to Tregs or the other way around that inducible overexpression of mutant *IKK2* can induce IL-17A expression in Tregs. This is of special relevance, as normally FoxP3 is known to repress IL-17A expression, and the authors show that FOXP3 Expression is not affected in those cells.

We agree with the reviewer that expression of IL-17A by Foxp3⁺ Tregs is remarkable. Indeed, the central findings of our manuscript are that *Ikkkb*^{G_oF} results in acquisition of an effector phenotype by Foxp3⁺ CD4⁺ T cells, that these cells contribute to pathology, and that this pathology resembles psoriasis and psoriatic arthritis.

The reviewer raises an interesting question of whether the IL-17A⁺ Tregs represent effectors that have upregulated Foxp3 or Tregs that have adopted effector function. The results of our mixed BM chimera experiment prove that *IKK2* acts cell autonomously to drive Treg expansion relative to WT cells in the same environment. Hypothesis-free analysis using single cell transcriptomics revealed the Tregs resemble a bona fide subset of Tregs reported previously rather than a T effector subset that has upregulated FoxP3.

In order to illuminate the question of precursor-progeny relation between Tregs and IL-17A⁺ Tregs further, we transferred GFP-negative T cells from *Ikkkb*^{mut/mut} × *Foxp3*^{GFP} mice into *Rag1*^{-/-} recipients but there was no induction of Foxp3 (GFP) after 8 weeks (Rebuttal Figure 3).

Rebuttal Figure 3. Sorted naive T cells (CD4⁺ CD44⁻ CD62L⁺ GFP⁻ cells) from *Ikkkb*^{WT} × *Foxp3*^{GFP} (top panel) or *Ikkkb*^{mut/mut} × *Foxp3*^{GFP} mice (lower panel) were injected intravenously into *Rag1*^{-/-} recipient mice (n=3 per genotype). At 8 weeks post-transfer, blood was collected from each mouse and analysed by flow cytometry. Donor CD4⁺ T cells are shown in Q1 and those that have upregulated Foxp3 (GFP) are shown in Q2.

We also investigated this question in vitro. Tregs (GFP⁺) and conventional T cells (GFP⁻) were isolated from *Ikkkb*^{mut/mut} × *Foxp3*^{GFP} mice and stimulated in vitro with anti-CD3 + antiCD28. Conventional T cells failed to upregulate Foxp3, whereas Tregs became more abundant. Thus, neither in vitro nor in vivo studies provide any evidence to support the idea that IL-17A⁺ Tregs observed in *Ikkkb*^{mut} mice are induced Tregs, or indeed, Th17 cells that have upregulated Foxp3.

Rebuttal Figure 4. Tregs (GFP⁺) and conventional T cells (GFP⁻) were isolated from *Ikkkb*^{mut/mut} × *Foxp3*^{GFP} mice, stimulated in vitro with anti-CD3+antiCD28, then harvested and analysed on day 3 for GFP expression.

Figure 5

The authors labeled Foxp3-positive Tregs with GFP and injected those cells into *IKK2*^{mut} or *IKK2*^{wt} mice on a *Rag1*^{-/-} background. While both mice show no signs of psoriasis-like dermatitis (at least until 21 weeks of age), injection of mutant *IKK2* Tregs into mutant *IKK2* mice induces the previously described phenotype. These results are important, however, they also imply several more open

questions which should be addressed:

- As IKK2 mutant Tregs can only induce the disease in IKK2 mutant animals, it seems that IKK2 overactivation in another cell type, such as keratinocytes, is important for disease onset. The authors should give some experimental insights into this. Does IKK2 mutation activate a keratinocyte or fibroblast response, that induces skin homing of IL17A-expression Tregs? This could be, at least, addressed *in vitro*.

As noted by the reviewer, Fig. 5 summarised results of a definitive experiment to determine whether there is a non-haematopoietic action of IKK2 during induction of pathology *in vivo*. As a result of this adoptive transfer experiment, we have striking *in vivo* evidence that the Foxp3⁺ cells arising as a result of *Ikk2*^{GoF} are necessary but not sufficient for pathology. The macroscopic and microscopic phenotypes reported after adoptive transfer of WT versus mutant cells provide evidence that non-haematopoietic cells act in concert with non-haematopoietic cells.

As a result of this experiment, non-haematopoietic cells in the skin must contribute to pathology, and there are several candidate skin cell types to account for this finding. As suggested by the reviewer, these include fibroblasts or keratinocytes, although additional skin-resident cell types could also contribute. Resolving which combination of receptor ligand pairs are acting to drive pathology will be an important but substantial task, and beyond the scope of the current study, which has focussed on the important question of how Tregs are modified by IKK2.

In response to the reviewer's question, however, we can provide some experimental insights based on additional *in vivo* analysis. Of course, a leading candidate is CD103, which is expressed on the modified Treg population. The ligand for CD103 (E-cadherin, *Cdh1*), however, is not differentially expressed in skin of mutant and WT mice. Despite this finding, we cannot discount the importance of the receptor ligand pair. Differential abundance of CD103⁺ Tregs in the presence of a fixed supply of E-cadherin might explain their recruitment, but would not easily explain the observation of a non-haematopoietic component to *Ikk2*^{GoF}-mediated pathology.

We also analysed skin transcriptomes for genes that fulfil the following conditions:

1. Differentially expressed between WT and mutant mice
2. Encode a receptor (or ligand) which is a DEG in skin but not lymphocytes, and for which the ligand (or receptor) is a DEG in NLTs.

This analysis identified several candidates. CXCL2 and CXCL3 both bind to CXCR2. CXCR2 is not expressed NLTs but is expressed in skin, and is NF- κ B regulated. CD69 is also highly expressed in NLTs but has numerous ligands, including S100A8/9 and these are minimally expressed in NLTs but highly expressed in skin. The results of this analysis are summarised in **Rebuttal Figure 5**.

Rebuttal Figure 5. Volcano plots of RNASeq data obtained from *Ikkkb^{mut/mut}* and *Ikkkb^{WT}* skin (ears and tail). Labels indicate expression of genes encoding proteins that emerge as candidates to account for recruitment of mutant NLTs.

- Where do IKK2 mutant Tregs localize upon injection? Do they home to the skin? The authors could easily test this by detecting the cells via GFP. If the cells do not home to the skin or joints, how could they induce psoriasis and arthritis then? If they are homing to the sites of inflammation, how? Which ligands-receptors might be involved? The results of such investigations should give more mechanistic insights how these Tregs can specifically induce skin and joint inflammation.

A central finding of our manuscript is that IL-17A+ Foxp3+ Tregs are located in sites of pathology, which we have shown by recovering these cells from skin and bone marrow (**Fig. 3J-R**). Candidate receptor ligand pairs are discussed in response to the previous question.

In general:

The method section lacks some fundamental information. It is not specified if the results are similar in female and male mice. The origin of antibodies is not described. Information on digestion enzymes, etc is missing.

Thank you. We have now revised and expanded in the methods

Reviewer #2 (Remarks to the Author):

*In their study Cardinez et al. dissect the phenotype resulting in mice with an *Ikkkb* gain-of-function mutation. Their analysis reveals dose dependent (mut vs het) inflammatory disease characteristic of psoriatic arthritis. Skin pathology is especially evident and occurs even in a proportion of het animals. Analysis of lesional and circulating immune populations indicates preferential expansion of Tregs, especially those with a tissue resident phenotypic and transcriptional signature. Surprisingly these Tregs also exhibit elevated IL-17 expression. The authors provide convincing evidence that it is these IL-17 producing Tregs that contributes to pathology. Overall, this is a robust study of broad interest that defines a potentially important role for IKK2 in Treg function. However, the mechanisms by which *Ikkkb* mutation produces this Treg activity are not clearly defined. I also have specific concerns about the single cell RNA-seq reference population used for analysis of skin resident Tregs. These concerns and suggestions for improvement are described below:*

1. The concept of Tregs contributing to inflammatory processes is not new and needs to be more fully discussed. Line 97 they cite an example of conversion but there are other contexts that have been described, especially for tissue resident Treg. ie cite and discuss: DOI: 10.1126/science.1155209; DOI: 10.1126/sciimmunol.abg2329

Thank you, we have now added this reference and additional comments to the introduction.

2. Is there any evidence of inflammation/pathology in trunk skin? Due to higher hair follicle density, there are typically more Tregs in these sites so if not discussion would be warranted.

Skin changes in the back occur later than those in tail and ears (**Rebuttal Figure 4; Supp Fig. 1B**), although ascertainment of back skin involvement is difficult because it often only becomes apparent after shaving the mice, and this is not a routine component of our assessment of extant mice under our current ethics protocol. We have now analysed involved back skin and confirmed that there is a similar enrichment for Tregs to that observed in tail and ears (see Response to reviewer 1).

Rebuttal Figure 6. Appearance of back skin from mice of indicated genotypes.

3. Given that pathology likely hinges on the absolute quantity of IL-17 produced by Tregs their cell number (particularly for Fig. 1K/skin; ie cells/g or cells/cm²) should be presented in addition to frequency. Also in Fig. 1K the representative flow plot does not show convincing Foxp3+ staining. The authors could perform more robust quantification with their Foxp3-GFP reporter cross.

We thank the reviewer for these suggestions. We have now added quantification of Tregs in the lesional skin (**Rebuttal Figure 7; Supp Fig 1C-E**) and updated the Foxp3 staining result.

Rebuttal Figure 7. Quantification of Foxp3+ cells from lesional skin

4. The mixed bone marrow chimera in Figure 3D-F is a compelling experiment but only showing 3 wt/het and 2 wt/hom mice is not sufficient to be convincing. The expansion is less than 2 fold in the hom so strong statistical analysis would be helpful.

In line with the reviewer's suggestion, we have repeated the mixed chimera experiment. The results from the repeat experiment do not change our conclusion but as a result of the increased numbers, the chance of a type I error is now very small. We have updated the figure describing these results (**Rebuttal Figure 8; Fig. 3D-F**).

Rebuttal Figure 8. Analysis of cell-intrinsic action of IKK2 by construction of mixed bone marrow chimeras. Irradiated *Rag1*^{-/-} mice were reconstituted with 50:50 mixtures of CD45-allotype marked donor bone marrow from *Ikkkb*^{WT}, *Ikkkb*^{+/-mut} or *Ikkkb*^{mut/mut} mice. Foxp3⁺ Tregs were analysed 8 weeks after reconstitution according to allotype of donor origin. Results are pooled from two identical experiments.

5. In Fig 4A the authors state there is no significant difference in their suppression assay. It actually looks like the mut/mut has better suppression. Please quantify to make this assertion.

Thank you. This information now been added as **Fig. 4B**.

6. The authors have demonstrated that the *Ikkkb* *gof* mutation leads to Treg expansion and simultaneous preferential expression of IL-17 but I'm unclear if and how these two Treg behavioral outcomes are linked. Is there evidence of enhanced survival or proliferation of the CD103⁺ Tregs with the *Ikkkb* mutation? Is IL-17 expression elevated compared to CD103⁺ wild type skin Tregs? Also does the *Ikkkb* *gof* lead to increased IL-17 production/secretion on a per cell basis?

The aim of our single cell experiments was to answer the question raised by the reviewer. We have delineated how IL-17A⁺ Tregs result from the selective expansion of subsets of Tregs, primarily NLTs. This is plausible, as even in WT mice, this Treg subset is characterised by relatively high expression of NF-κB genes. Further, our evidence suggests that under the influence of *Ikkkb*^{Gof}, NLT skin adopt an inflammatory phenotype. Thus, our evidence indicates that Treg expansion and preferential IL-17A expression results from enhanced differentiation and NLTs. These results are summarised in Fig. 6-7 of the manuscript. We have now complemented this finding with analysis of Tregs from mutant and WT mice for Ki67 expression, which reveals evidence for increased proliferation. This result is shown below (**Rebuttal Figure 9**) and included in the revised manuscript (**Supp Fig. 3M-N**).

Rebuttal Figure 9. Analysis of recent cell proliferation of Tregs based on Ki67 expression.

7. *scRNA-seq* is performed on splenic and bone marrow Tregs from mutant mice and analysis and compared to the dataset in reference 28. From this it is concluded there is an expansion of Skin NLT. GSEA is used to identify enrichment of TNF-signaling/NF- κ B pathway in these Skin NLT. While ref 28 does a good job broadly identifying Treg cell states this data set actually only contains 154 skin derived Tregs. This is too small/biased to make reliable conclusions for the types of analyses performed here. Expand analyses with skin specific datasets. For example: DOI: 10.1126/sciimmunol.abg2329.

We thank the reviewer for drawing our attention to this important and relevant resource. We have now reanalysed our data according to the genes identified in DOI: 10.1126/sciimmunol.abg2329 and included these results (**Rebuttal Fig. 10; Supp Fig. 8**), accompanied by the following text to the results section:

Tregs in the skin have also been linked to elevated expression of TGF- β and integrin pathway genes⁵⁰. Similarly, our identified skin NLTs showed a significant increase of TGF- β pathway genes *Tgfb1*, *Itgav*, and *Itgb8* compared with other Treg clusters (**Fig. S8C**). NLT skin Treg also revealed significant enrichment of genes associated with TGF- β signalling. However, enrichment of genes associated with integrin pathways was not significant (**Fig. S8D**).

This suggested analysis provides additional support for our conclusion that *Ikk2*^{GoF} drives expansion of NLTs.

Rebuttal Figure 10.

Gene set enrichment analysis plots for the indicated TGF- β or integrin gene sets in the NLT skin cluster. HALLMARK_TGF_BETA_SIGNALING_NLT skin NES = 1.6922, padj = 0.0133. BIOCARTA_INTEGRIN_PATHWAY_NLT skin NES = -0.3308, padj = 0.7911. GOBP_INTEGRIN_MEDIATED_SIGNALING_PATHWAY_NLT skin NES = 0.3943, padj = 0.6950.

Rebuttal references

- 1 Skroza undefined N, Proietti undefined I, Bernardini undefined N, *et al.* Il-17 and Its Role in Psoriasis, Hidradenitis Suppurativa And Acne. *Internal Medicine and Care.* 2017;1:1–6.
- 2 Zwicky P, Unger S, Becher B. Targeting interleukin-17 in chronic inflammatory disease: A clinical perspective. *J Exp Med.* 2019;217:e20191123.
- 3 He H, Bissonnette R, Wu J, *et al.* Tape strips detect distinct immune and barrier profiles in atopic dermatitis and psoriasis. *J Allergy Clin Immunol.* 2021;147:199–212.

REVIEWERS' COMMENTS

Reviewer #1 (Remarks to the Author):

The authors sufficiently addressed all raised points by the reviewers. As already stated before, the manuscript shows important and interesting new data, on how a gain-of-function mutation, in this case, *Ikbkb*, can cause psoriasis and related arthritis. I recommend to accept the manuscript for publication. However, there are some minor things which still need to be corrected before publication:

Figure 1F: The photo is too dark

Figure 1H: Scale bars are missing, please add

Figure 1K: please indicate in the figure legend or method section how many animals/samples were used for bulk sequencing. In the volcano plot, variation within single samples/mice cannot be seen. Therefore, a differential gene expression table showing the relative counts of each gene per sample would be helpful to see the variation within each group, instead of just showing the summary data of the analysis. [At least for genes that are significantly regulated it would be very helpful] (as supplementary tables).

Figure 4D: A presentation of the data as single points per animal (instead of the mean) would be better to assess the differences in the body mass within each group, as the standard deviation seems to be quite high

Data availability: Disposition of the sequencing data to public databases and/or reference to the sequencing raw data (GSE Assession numbers) is missing

Method section: There are a lot of typing errors; please correct them. μL not ul (see line 839 or 840), $^{\circ}\text{C}$ not degree or C (see line 873, 860...), xg not g or G , still some agent concentrations or antibody clone IDs are missing

Reviewer #2 (Remarks to the Author):

The authors have fully addressed all of the concerns highlighted in my initial review. I believe their current manuscript is robust and of high significance to the field.

Point-by-point response to reviewers

Reviewer #1 (Remarks to the Author):

The authors sufficiently addressed all raised points by the reviewers. As already stated before, the manuscript shows important and interesting new data, on how a gain-of-function mutation, in this case, *Ikbkb*, can cause psoriasis and related arthritis. I recommend to accept the manuscript for publication. However, there are some minor things which still need to be corrected before publication:

Figure 1F: The photo is too dark

Corrected

Figure 1H: Scale bars are missing, please add

Added

Figure 1K: please indicate in the figure legend or method section how many animals/samples were used for bulk sequencing. In the volcano plot, variation within single samples/mice cannot be seen. Therefore, a differential gene expression table showing the relative counts of each gene per sample would be helpful to see the variation within each group, instead of just showing the summary data of the analysis. [At least for genes that are significantly regulated it would be very helpful] (as supplementary tables).

We have added the requested information to the legend and now included a supplementary table that includes the gene expression data for each individual mouse.

Figure 4D: A presentation of the data as single points per animal (instead of the mean) would be better to assess the differences in the body mass within each group, as the standard deviation seems to be quite high

We have changed the format of the figure as requested.

Data availability: Disposition of the sequencing data to public databases and/or reference to the sequencing raw data (GSE Assession numbers) is missing

I think the reviewer missed this as it was already included.

Method section: There are a lot of typing errors; please correct them. μ L not ul (see line 839 or 840), °C not degree or C (see line 873, 860...), xg not g or G, still some agent concentrations or antibody clone IDs are missing

The methods section has been corrected

Reviewer #2 (Remarks to the Author):

The authors have fully addressed all of the concerns highlighted in my initial review. I believe their current manuscript is robust and of high significance to the field.